# How weather events modify aerosol particle size distributions in the Amazon boundary layer

Luiz A. T. Machado[1,2], Marco A. Franco[2], Leslie A. Kremper[1], Florian Ditas[1,a], Meinrat O. Andreae[1,3,4], Paulo Artaxo[2], Micael A. Cecchini[2], Bruna A. Holanda[1], Mira L. Pöhlker[1], Ivan Saraiva[5], Stefan Wolff[1], Ulrich Pöschl[1], and Christopher Pöhlker[1]

[1]Multiphase Chemistry Department, Max Planck Institute for Chemistry, Mainz, Germany
[2]Instituto de Física, Universidade de São Paulo, São Paulo, Brazil
[3]Scripps Institution of Oceanography, University of California San Diego, La Jolla, CA 92037, USA
[4]Department of Geology and Geophysics, King Saud University, Riyadh, Saudi Arabia
[5]Sistema de Proteção da Amazônia, Manaus, Brazil
[a]now at: Hessian Agency for Nature Conservation, Environment and Geology, 65203 Wiesbaden, Germany

**Correspondence:** Luiz A. T. Machado (l.machado@mpic.de)

**Abstract.** This study evaluates the effect of weather events on the aerosol particle size distribution (PSD) at the Amazon Tall Tower Observatory (ATTO). This research combines in-situ measurements of PSD and remote sensing data of lightning density, brightness temperature, cloud top height, cloud liquid water, and rain rate and vertical velocity. Measurements were obtained by a scanning mobility particle sizers (SMPS), the new generation of GOES satellites (GOES-16), the SIPAM S-band radar, and the LAP 3000 radar wind profiler recently installed at the ATTO-Campina site. The combined data allow exploring changes in PSD due to different meteorological processes. The average diurnal cycle shows a higher abundance of ultrafine particles ($N_{UFP}$) in the early morning, which is coupled with relative lower concentrations in Aitken ($N_{AIT}$) and accumulation ($N_{ACC}$) mode particles. From the early morning to the middle of the afternoon, an inverse behavior is observed, where $N_{UFP}$ decreases and $N_{AIT}$ and $N_{ACC}$ increase, reflecting a typical particle growth process. Composite figures show an increase of $N_{UFP}$ before, during, and after lightning was detected by the satellite above ATTO. These findings strongly indicate a close relationship between vertical transport and deep convective clouds. Lightning density is connected to a large increase in $N_{UFP}$, beginning approximately 100 minutes before the maximum lightning density and reaching peak values around 200 minutes later. In addition, the removal of $N_{ACC}$ by convective transport was found. Both the increase in $N_{UFP}$ and the decrease in $N_{ACC}$ appear in parallel with the increasing intensity of lightning activity. The $N_{UFP}$ increases exponentially with the thunderstorm intensity. In contrast, $N_{AIT}$ and $N_{ACC}$ show a different behavior, decreasing from approximately 100 minutes before the maximum lightning activity and reaching a minimum at the time of maximum lightning activity. The effect of cloud top height, cloud liquid water, and rain rate shows the same behavior, but with different patterns between seasons. The convective processes do not occur continually but are probably modulated by gravity waves in the range of 1 to 5 hours, creating a complex mechanism of interaction with a succession of updrafts and downdrafts, clouds and clear sky situations.

The radar wind profiler measured the vertical distribution of the vertical velocity. These profiles show that downdrafts are mainly located below 10 km, while aircraft observations during the ACRIDICON-CHUVA campaign had shown maximum

concentrations of ultrafine particles mainly above 10 km. Our study opens new scientific questions to be evaluated in order to understand the intricate physical and chemical mechanisms involved in the production of new particles in Amazonia.

# 1    Introduction

The Fifth Assessment Report of the Intergovernmental Panel on Climate Change (AR5 of IPCC, 2013) shows that clouds and aerosols contribute the largest uncertainty in our understanding of the Earth's changing energy budget. Aerosols impact climate, clouds, and precipitation by the direct and indirect effects. Studies based on observations and modeling have improved knowledge and reduced uncertainties regarding the aerosol direct effect (e.g., Ramanathan et al., 2001; Hansen et al., 1997); see also Haywood and Boucher (2000) for a review of the estimated direct global annual mean radiative forcing. The effects of natural and anthropogenic sources, such as forest fires and urban pollution, on aerosol-cloud-precipitation interactions, including cloud invigoration as well as precipitation reduction by increased aerosol loading and the associated effects on cloud microphysics have been intensely studied (e.g., Albrecht, 1989; Andreae et al., 2004; Rosenfeld, 2018; Twomey and Warner, 1967; Rosenfeld, 1999; Koren et al., 2005; Rosenfeld et al., 2008; Cecchini et al., 2017a; Heikenfeld et al., 2019). In addition, the knowledge about aerosol characteristics, cloud condensation nuclei (CCN), secondary organic aerosol (SOA), and volatile organic compounds (VOCs) has been greatly improved, specifically in the Amazon region (e.g., Pöschl et al., 2010; Pöhlker et al., 2016, 2018; Palm et al., 2018; Yáñez-Serrano et al., 2020; Holanda et al., 2020; Saturno et al., 2018; Leppla et al., 2021; Schrod et al., 2020). Although these studies have advanced our knowledge of aerosol-cloud interactions, only very few attempts have been made to describe how clouds modify the aerosol properties.

For the Amazon, this topic was recently addressed by two field experiments: Observations and Modeling of the Green Ocean Amazon (GoAmazon2014/5) (Martin et al., 2016, 2017) and ACRIDICON-CHUVA[1] (Wendisch et al., 2016; Cecchini et al., 2017b; Machado et al., 2018). Gerken et al. (2016) discussed the downward transport of ozone by convective clouds and used the surface equivalent potential temperature ($\theta_e$) as a proxy of convective events that produced an increase in ozone concentration at the surface during rainfall events. Wang et al. (2016) analyzed the origin of small aerosol particles in the boundary layer (BL). They found high concentrations of these particles in the free troposphere up to 6 km and concluded that they were transported down by convective downdrafts during intense precipitation events. Andreae et al. (2018) showed enhanced concentrations of ultrafine particles in the upper troposphere, which were much higher than in the planetary BL. They suggested that these particles were produced in the outflow from deep convective events and were returned to the surface by convective downdrafts. Williamson et al. (2019), using aircraft measurements over the Pacific and Atlantic Oceans also observed new particle formation in the tropical upper troposphere and suggested that these particles descend towards the lower troposphere, where they can act as CCN, modifying cloud properties.

This study evaluates the effect of weather events on particle number size distribution (PSD) changes, combining satellite and radar data with measurements from two scanning mobility particle sizers (SMPS) at 60 and 325 m height at the Amazon Tall

---

[1]Aerosol, Cloud, Precipitation, and Radiation Interactions and Dynamics of Convective Cloud Systems–Cloud Processes of the Main Precipitation Systems in Brazil: A Contribution to Cloud Resolving Modeling and to the GPM

Tower Observatory (ATTO) (Andreae et al., 2015). The PSD data is analyzed for the main cycles and cloud characteristics in the Central Amazonian region, highlighting the relationship between lightning storms and changes in PSD. This study explores changes in PSD by the atmosphere's seasonal and diurnal cycles and intradiurnal variability, and evaluates the relationships between PSD and lightning, rainfall, cloud top height, cloud liquid water, and brightness temperature during the dry and wet seasons. These analyses will help to understand the new particle formation observed during convective events.

## 2 Data and Methods

This study combines different types of datasets, all colocated at the ATTO site, which has been described in detail by Andreae et al. (2015). The data were obtained from the new generation of GOES satellites (GOES-16), the SIPAM S-band radar, two SMPS instruments sampling from 60 and 325 m height above ground, and the LAP 3000 radar wind profiler installed at the ATTO-Campina site. ATTO-Campina is a new site to complement ATTO measurements with cloud physics instruments, specifically designed for the study of aerosol-cloud-precipitation interactions. The Campina site is located about 4 km from the main tower in a small isolated clearing, characterized by relatively dry and sandy soils as well as the typical low Campina vegetation type (Junk et al., 2011). The small size of the trees allowed the installation of remote sensing instruments to measure cloud physics. All datasets used in this study were organized and compiled to produce two years of data with 10-minute time resolution, from October 2017 to September 2020. Detailed descriptions of the individual datasets are presented below.

### 2.1 Scanning Mobility Particle Sizers

This study focuses on particle number size distribution measurements obtained from two Scanning Mobility Particle Sizers (SMPS, TSI Inc., Shoreview, USA), each comprising a differential mobility analyzer (DMA) and condensation particle counter (CPC). Both SMPS systems were measuring in parallel at two different sampling heights at the ATTO site. One SMPS was sampling from the 60 m inlet of the triangular mast and the second from the 325 m inlet of the Tall Tower. Due to several instrument failures in the course of the long-term measurements, the DMA and CPC instruments were replaced as necessary. These replacements involved different DMA and CPC models: the measurements at the 60 m inlet involved the classifier models 3080 and 3082, the DMA model 3081, and the CPC model 3722). The measurements at the 325 m inlet involved the classifier model 3082, the DMA model 3081, and the CPC models 3722 and 3750 (for details, see Andreae et al., 2015; Pöhlker et al., 2016). Both instruments are located in air-conditioned laboratory containers at the foot of the towers. Sample air was transported through stainless steel tubes (finetron tubes, Dockweiler AG, Neustadt-Glewe, Germany) and dried to a relative humidity (RH) below 40 %. At the 60 m inlet, an automatic regenerating silica gel adsorption aerosol dryer, as described in Tuch et al. (2009), was installed upstream of the instruments since 2014 and was replaced by a custom-built and automated condensation aerosol dryer in March 2020. At the 325-m inlet, an automated condensation drier was used since the beginning of measurements. Both SMPS instruments cover a particle size range from 10 to 400 nm, and the sizing accuracy was frequently checked using monodisperse polystyrene latex particles. Data were acquired and exported with the Aerosol Instrument Manager Software (AIM, Version 9 & 10, TSI Inc., Shoreview, USA). The original temporal resolution was 5

minutes and was integrated to 10 minutes to maintain the same time resolution as the other data. The data were adjusted for standard temperature (273.15 K) and pressure (1013.25 hPa). The theoretical losses due to the long inlets at 60 m and 325 m were corrected based on a methodology using size-dependent correction factors, as described by von der Weiden et al. (2009).

For the purpose of our analysis, we divided the PSD, obtained by the SMPS over the range from 10 to 400 nm, into the following three individual size classes: (a) ultrafine aerosol particles (UFP) from 10 to 50 nm, (b) Aitken mode particles from 50 to 100 nm and (c) accumulation mode particles from 100 to 400 nm (see also multimodal fitting in Pöhlker et al., 2016). In order to evaluate the particle characteristics along the classes, we computed the cross-correlation between the different particles sizes for the Central Amazonian dry (Jun to Oct) and wet (Dec to Apr) seasons. Figure 1 shows how well the multimodal PSDs are described by the aforementioned modes by means of a cross-correlation plot for the dry season. The accumulation mode particles and Aitken mode particles have a clearly defined separation at 100 nm, mainly during the wet season, which is the location of the Hoppel minimum (Hoppel et al., 1996) as observed in Central Amazonia by Pöhlker et al. (2016). However, the separation between Aitken mode particles and UFP is not clearly evident. The Aitken mode particles have a smaller cross-correlation between the particles, which can likely be explained by the fact that this range of particles is in the transition between the other two modes. Specifically, the Aitken mode particles appear to have sometimes nucleation mode particle properties and sometimes accumulation mode particles properties. Clearly, the UFP and accumulation mode particles are well cross-correlated between the different particle sizes; however, for the wet season this separation is not well defined. In the wet season there is an increase in cross-correlation from Aitken particles to UFP, but the cross correlation inside the population of the UFP is smaller than during the dry season. The UFP are in a size range where Aitken and nucleation particles modes overlap, therefore we adopted the ultrafine particle denomination for this size range, which showed to be very sensitive and with a different behavior from the other particle classes, as will be shown in this study. Fan et al. (2018) used the same size range to show the effect of UFP in the strengthening of convective updrafts. Given the overlap between the nucleation and Aitken modes and the growth of particles from one mode into the other, the cut at 50 nm is somewhat arbitrary, but as will be shown below, provides a practical basis for discussing the behavior of these particle classes in response to meteorological events and across diurnal and seasonal time scales. Table 1 summarizes the definitions, abbreviations and size ranges of aerosol particle classes used in this study.

**Table 1.** Definitions, name abbreviations, and symbols of the three aerosol particle size ranges used in this study.

| Name of aerosol particle class | Concentration | Diameter ($D$) range [nm] |
|---|---|---|
| Ultrafine particles (UFP) | $N_{\text{UFP}}$ | $10 < D \leq 50$ |
| Aitken mode particles | $N_{\text{AIT}}$ | $50 < D \leq 100$ |
| Accumulation mode particles | $N_{\text{ACC}}$ | $100 < D \leq 400$ |

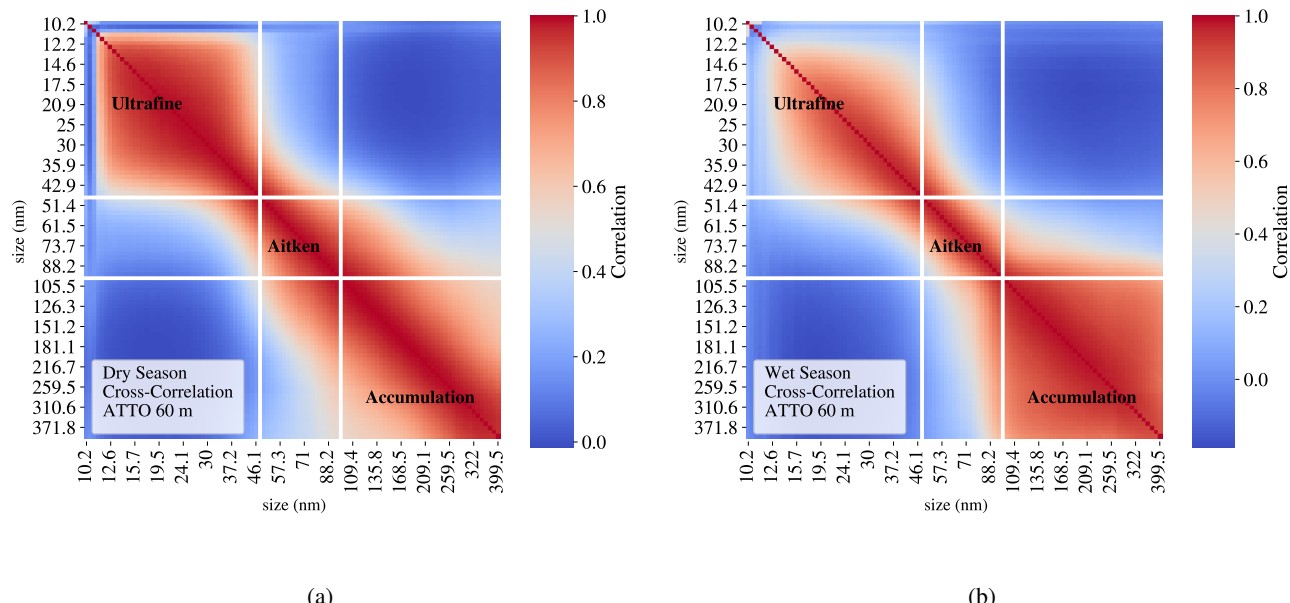

**Figure 1.** Particle number size distribution cross-correlation sampled at 60 m height at ATTO. a) Dry season (Jun to Oct) and b) wet season (Dec to Apr). The cross-correlation underlines the distinction into ultrafine, Aitken, and accumulation modes.

## 2.2 Geostationary Operational Environmental Satellite

The Geostationary Operational Environmental Satellite (GOES) is the operational NOAA (National Oceanic and Atmospheric Administration) satellite covering South America. In 2016, NOAA launched the new generation of geostationary satellites, GOES-16, located at 75.2 W degrees. GOES-16 has a new set of instruments, in particular, used in this study, are the Advanced Baseline Imager (ABI) and the Geostationary Lightning Mapper (GLM). The GOES-16 ABI sensor data employed are the ABI channel 13 (10.35 μm). The ABI channel 13 has 2 km and the GLM has 8 km spatial resolution, both of them in their original satellite projection. Channel 13 is usually used to estimate the convective cloud depth, where lower brightness temperature ($T_{IR}$) relates to higher cloud tops simply because the temperature decreases with height in the troposphere (Adler and Mack, 1986). From ABI channel 13, three variables were selected, namely, the ATTO colocated $T_{IR}$ as well as the frequency of occurrence of $T_{IR} < 284$ K and $T_{IR} < 245$ K, in a grid of 5 by 5 pixels centered on ATTO. Clear sky radiance in Amazonia is associated with the mixed layer brightness temperature due to the high amount of water vapor near the surface. Machado et al. (2002) used the 284 K threshold to estimate the total cloud cover. The threshold of 245 K is usually employed to describe clouds associated with convective events in tropical regions.

The lightning data were obtained from the GLM, which detects lightning from a 0.74 μm CCD array with a very high temporal resolution, so it can compare the energy peak at different times to define whether it is a lightning event or not (detailed

description in Goodman et al., 2013). To obtain a better representation of lightning around ATTO, the GLM information was computed as the lightning density: the number of events every ten minutes in an area of 5 by 5 GLM pixels, which corresponds to a description of lightning activity in a 25 km radius.

## 2.3 S-Band Radar

The SIPAM S-band radar is a weather radar, located in Manaus and operated by SIPAM – (Sistema de Proteção da Amazônia).
This S-band radar (10 cm wavelength – 3.0 GHz) has a beam width of 1.8° and a 240 km radius coverage. The radar run occurs every 12 minutes, a volume scan strategy covering 17 elevations. The volume scan produces a 3-D description of the atmosphere. The following parameters were obtained from the 3-D field: The constant altitude plan position indicator (CAPPI) is the reflectivity (dBZ) field at a specific height, in this case we have used the CAPPI at 3 km. There is a direct relationship between reflectivity and rainfall; the Marshall-Palmer relationship assumes an exponential drop-size distribution and defines
rain rate ($R$) as: $Z = 200R^{1.6}$, where $Z$ is the reflectivity in ($mm^6 m^{-3}$) and $R$ is in $mm\,h^{-1}$. The CAPPI 3 km is usually associated with the rain rate if rainfall evaporation below 3 km is neglected. The 3 km height is employed to be free of clutter and the Earth curvature effects in the 240 km radius. The echo top corresponds to the maximum height of cloud threshold of 20 dBZ and can be considered as the height of the top of the rain rate associated with the reflectivities describe above. The 20 dBZ echo top is approximately the height of the cloud top with rain drops corresponding approximately to the rain rate of
1 $mm\,h^{-1}$. The VIL is the vertically integrated liquid ($kg\,m^{-2}$). VIL is obtained by the vertical integration of the reflectivity as described by Greene and Clark (1972).

## 2.4 LAP3000

The LAP3000 radar wind profiler (RWP) is a vertically pointing radar capable of measuring Doppler velocity spectra profiles under both clear sky and precipitation conditions. It operates in the 1290 MHz frequency range, with a 7.1° beam width, and
145 obtains the Doppler spectra based on oscillations of the refractive index of air in the case of clear sky or based on droplet relative motion in the case of precipitation. The instrument is set up to operate in three different modes, one optimized for clear sky and the other two for precipitation. The precipitation modes are obtained in a similar fashion as in Tridon et al. (2013), which aims to convert the RWP to an S-band-like profiler. A similar approach has already been used in the Amazon during the two-year deployment of the Atmospheric Radiation Measurement (ARM) Mobile Facility (AMF) during the GoA-
150 mazon2014/5 experiment (Martin et al., 2017, 2016; Giangrande et al., 2017). The present study only used data from one of the precipitation modes, which is able to sample between 1 and 15 km altitude with spatial and temporal resolutions of 210 m and 9.5 s, respectively. The pulse repetition frequency (PRF) is set to 8300 Hz, and the velocity range is between -20 $m\,s^{-1}$ and +20 $m\,s^{-1}$ with 0.1 $m\,s^{-1}$ resolution. The spectra moments are integrated into 5-min intervals for the analysis of the up- and downdrafts.

This instrument was employed to compute the statistical profile of the vertical velocity inside convective clouds. The analysis of resultant vertical velocity obtained by the RWP at the ATTO-Campina site was done by computing the contoured frequency-by-altitude diagram, a tool normally used in radar data analysis (detailed description in Yuter and Houze, 1995). As this is a

new instrument installed at ATTO site, the calculation was performed for only 54 days of measurements between 16 October and 8 December 2020. The vertical resolution of the contoured frequency-by-altitude diagram is set to 840 m, while the vertical velocity resolution is $0.1\,\mathrm{m\,s^{-1}}$.

## 3 Results

This section explores how particle size distribution changes as a function of the main atmospheric cycles and cloud cover characteristics. The main atmospheric cycles in Amazonian are the seasonal and diurnal cycles. But, the atmosphere also varies in the intradiurnal and interdiurnal frequencies. This section will also evaluate how the intradiurnal variability is associated with the PSD. Intradiurnal cloud variability is mainly forced by gravity waves, and this topic will be discussed in subsection 3.7. The PSD associated with the cloud characteristics will be evaluated mainly by looking at the lightning density as a proxy of deep convection. This study also evaluates PSD as a function of cloud top height, rainfall, cloud-integrated liquid water, and brightness temperature. These evaluations allow building a holistic view of how weather modifies the particle size distribution. One of the main topics covered in this study is the association between BL dynamiscs and new particle formation. One remarkable feature already observed in case studies is the increased concentration of UFP during deep convective events. Section 4 will discuss and evaluate the different possibilities and the associated processes.

### 3.1 Seasonal Cycle

The seasonal cycle in the Amazonian region is very important to be studied because this region presents a very well-defined annual variation, with a dry season in the austral winter and a wet one in the austral summer. Moreover, the transition phases between wet to dry and dry to wet seasons present distinct behaviors of chemical and atmospheric properties. The dry season has higher particle concentration and the most intense thunderstorms. In contrast, the wet season has higher precipitation accumulation but with lower rain rates and particle concentration. Machado et al. (2004) and Pöhlker et al. (2018) present a completely description of the seasonal variability. The total particle number concentration varies by about one order of magnitude between wet and dry seasons (Andreae et al., 2015; Andreae, 2009). Pöhlker et al. (2016) discuss the seasonal variation of cloud condensation nuclei (CCN) in Central Amazonia and conclude that the variability in the CCN concentrations is mostly driven by the aerosol particle number concentration and size distribution. Moran-Zuloaga et al. (2018) present multi-year accumulation and coarse mode measurements at ATTO and discuss the large diversity of the aerosol sources. Varanda Rizzo et al. (2018) present a multi-year analysis of submicrometer particle size spectra at ATTO for the wet and dry season. Figure 2 clearly shows this high variability for all sizes, with a maximum number concentration at the end of the dry season, mainly in the range between 100 and 200 nm.

In addition, Fig. 2 shows the seasonal evolution of the brightness temperature ($\mathrm{T}_{IR}$) and the lightning density. The brightness temperature starts to increase from May to July, characterizing the decrease in cloud cover and the frequency of occurrence of convective clouds, followed by the increase in particle number concentration. High $\mathrm{T}_{IR}$ values ($\mathrm{T}_{IR} > 284\,\mathrm{K}$) are associated with low cloud cover or shallow convection and cold $\mathrm{T}_{IR}$ ($\mathrm{T}_{IR} < 235\,\mathrm{K}$) are associated with deep convection. April is the

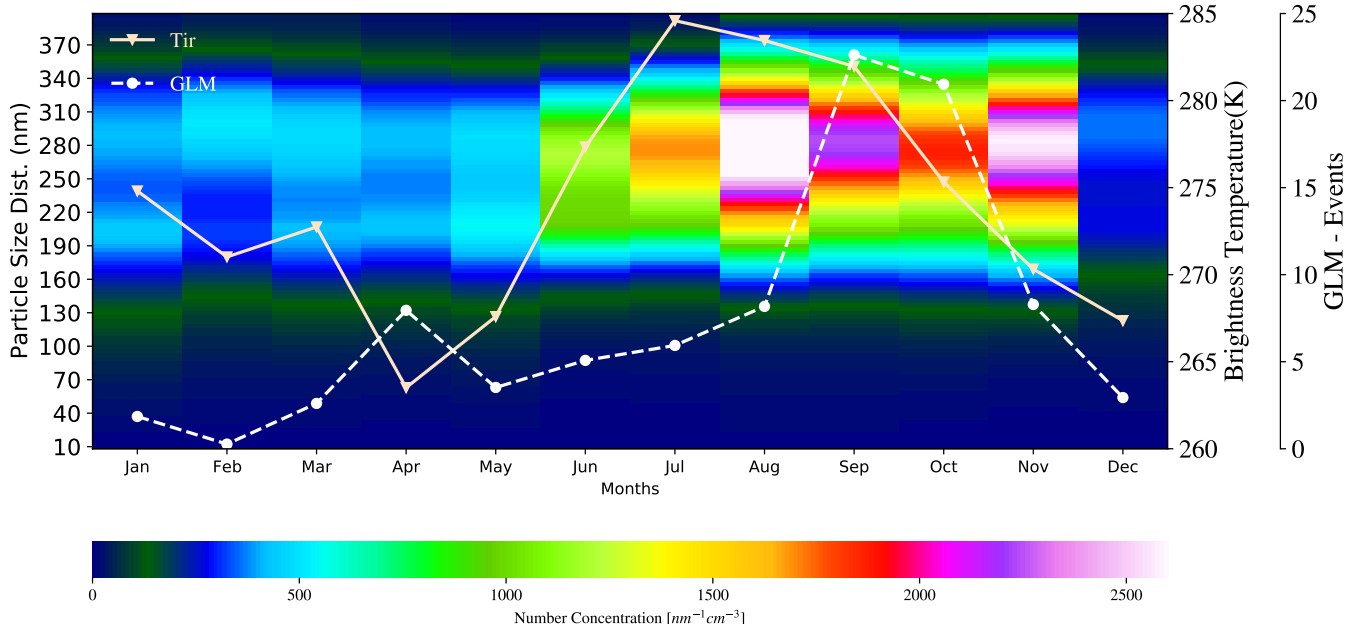

**Figure 2.** Seasonality of particle size distributions at ATTO (60 m sampling height), brightness temperature ($T_{IR}$), and lightning density (number of lightnings per 5 minute time bins in area of $50 \times 50 \, km^2$ around ATTO). All values are shown as monthly means. Average time frame for all parameters ranges from Oct 2018 to Sep 2020.

month with the coldest $T_{IR}$ and the rainiest month, but the maximum lightning activity is in September when deep convective rainfall events become more frequent. The brightness temperature and PSD relationship is straightforward, where higher $T_{IR}$ is associated with reduced rain and higher particle concentration during the dry season and lower particle concentration during the wet season when $T_{IR}$ is colder, representing higher (colder) cloud tops. However, the lightning activity shows the sharpest increase at the time when the aerosols have the maximum concentration (August). The transition between dry to wet season is the time when clouds are deeper and more intense, consequently having more electrification activity.

The seasonal cycle of the particle number concentration at different altitudes shows interesting features. From the wet to the dry season, the $N_{UFP}$ is nearly constant at 60 m, but increases at 325 m (Figure 3a). The absolute value of the difference between the concentration at 60 m and 325 m is small and could depend on the correction employed for losses in the long inlet tubing at 325 m. The $N_{ACC}$ (Figure 3b) has the same phase and nearly the same average concentration at 60 m and 325 m (511 and 556 $cm^{-3}$, respectively).

### 3.2 Diurnal Cycle

Clouds in Amazonia have a typical diurnal cycle during both the wet and dry seasons (Machado et al., 2004). The cloud diurnal cycle as well as BL processes modify the PSD (Betts et al., 2002; Saturno et al., 2018). Figure 4 shows the diurnal cycle of the PSD for the dry and wet seasons. Although the dry season has a much larger aerosol concentration than the wet season – on

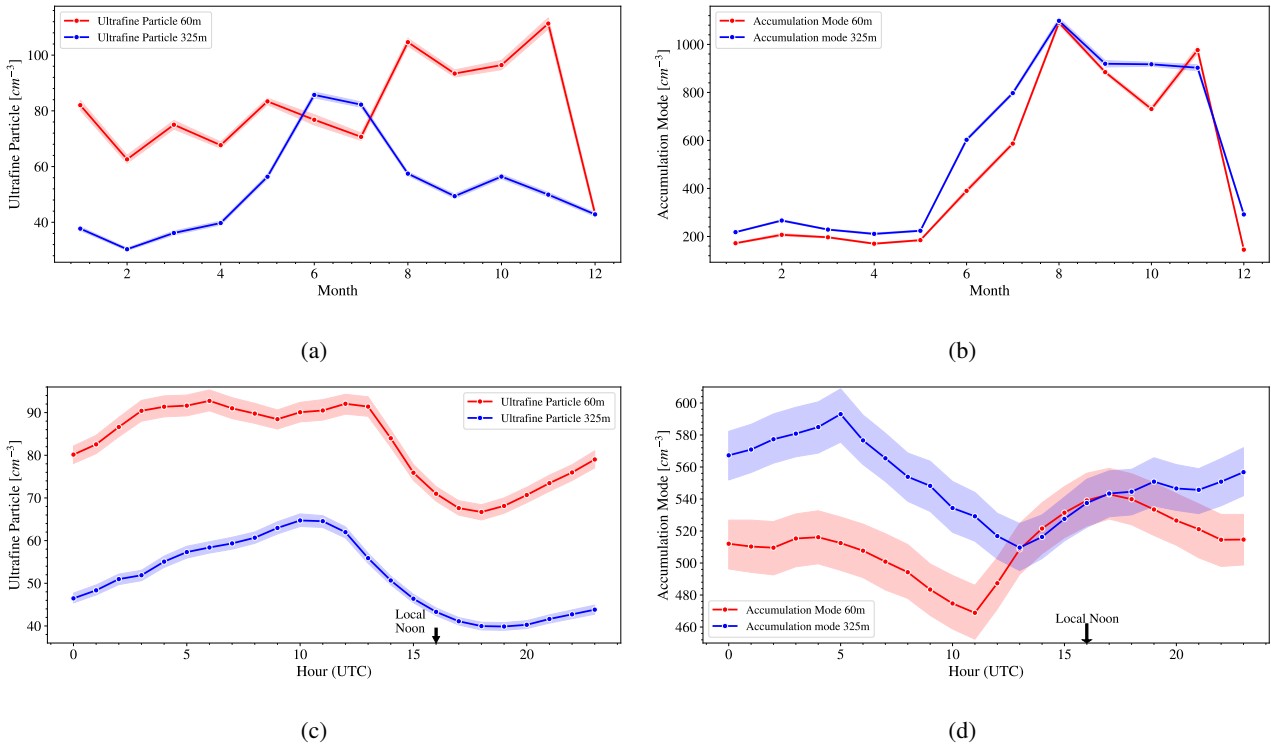

**Figure 3.** Seasonal and Diurnal variation of the number concentration of ultrafine and accumulation mode particles: a) Monthly average of the ultrafine particles at 60 m and 325 m, b) Monthly average of the accumulation mode at 60 m and 325 m, c) Hourly average of ultrafine particles at 60 m and 325 m, d) Hourly average of accumulation mode at 60 m and 325 m. The shaded area corresponds to a confidence interval of 95 %. Time in UTC, LST is UTC-4hours

average about factor 5 higher (Moran-Zuloaga et al., 2018) – the diurnal cycle for the wet and dry seasons commonly shows a minimum in $N_{\mathrm{ACC}}$ and $N_{\mathrm{AIT}}$ at the end of the night and during early morning (i.e., 04:00 to 08:00 local time (LST), which is UTC -4 h). It affects mainly the smaller sizes in the accumulation mode. This minimum during late night and early morning is followed by an increase in $N_{\mathrm{ACC}}$ up to the time when $T_{IR}$ starts do decrease and lightning density increases in the afternoon. This is the time when the maximum in $N_{\mathrm{ACC}}$ is observed. Moran-Zuloaga et al. (2018) show, for the wet season, that episodic
intrusions of African long-range transport of aerosols strongly increase the coarse mode in the early afternoon, when air is mixed down into the PBL from the dust-laden layer aloft.

The increase in the number concentration between morning and noon is coincident with the time when surface sensible heat flux becomes positive, the nocturnal BL is eroded, and the convective BL is rapidly built (Stull, 1988; Angevine et al., 2020; Fisch et al., 2004). At this time, the BL evolves from the stable stage (nocturnal BL) to the convective BL (mixed
layer). Following Stull (1988) classification of the stages of BL evolution, Henkes et al. (2021) show in detail the BL height evolution and surface fluxes for these stages of the development of the convective BL in Central Amazonian region during the dry season. The growing stage begins at sunrise and is determined by the time when the surface heat flux passes above zero

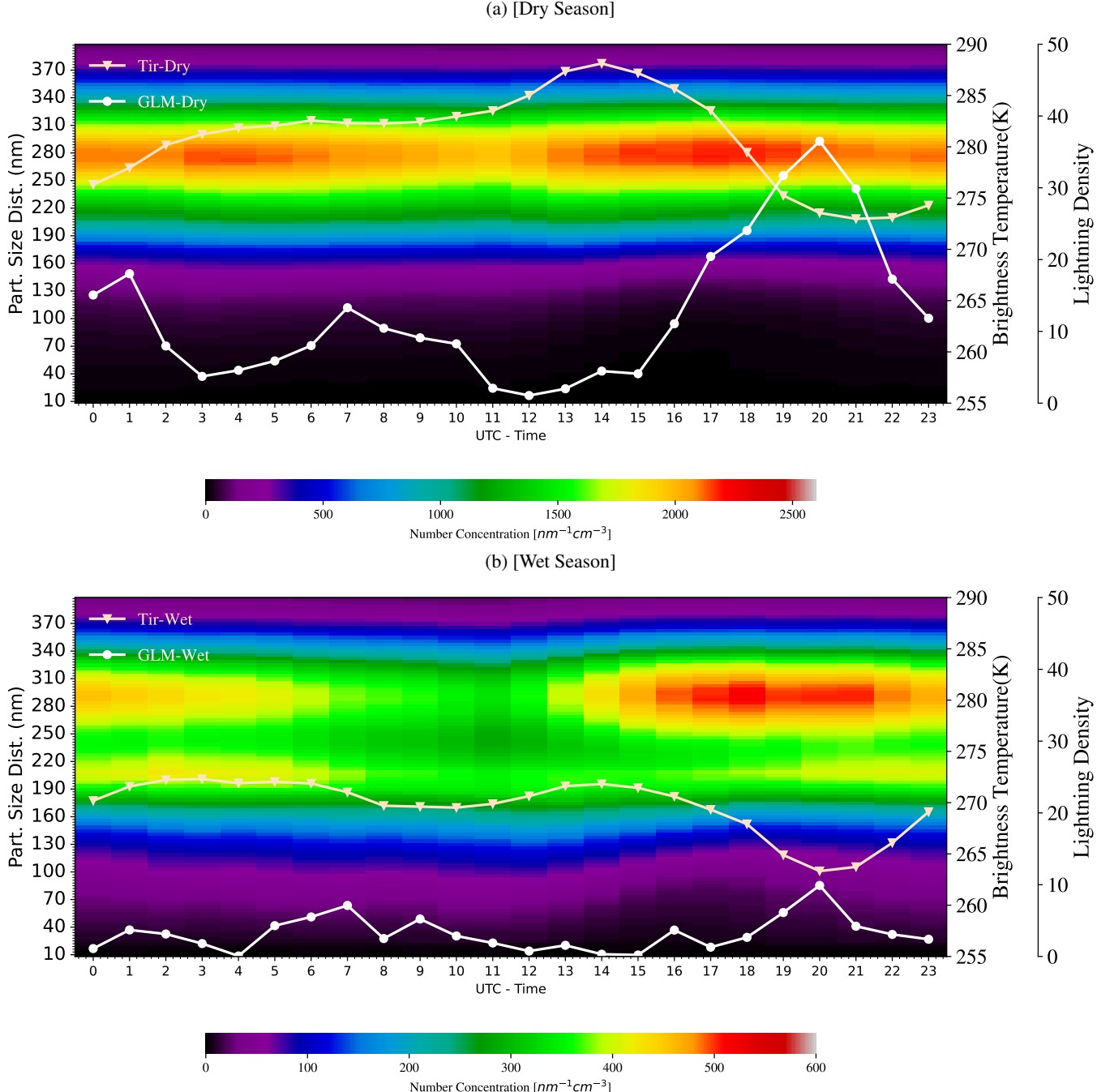

**Figure 4.** Diurnal cycle of the particle size distribution at ATTO (60 m sampling height) during the dry season (Jun-Oct) [a] and wet season (Dec-Apr) [b]. The monthly mean brightness temperature and lightning density (number of lightning events in 5 minutes in 50x50 km$^2$) are plotted as white and gray lines with solid lines. Time is UTC, LST is UTC -4 hours

and the turbulent kinetic energy promotes rapid growth of the convective BL. At this time, the BL could exports particles to the free troposphere (the Earth's atmosphere above the planetary BL) or/and mix from free atmosphere into the lower BL; Later in the afternoon, they are removed by precipitation scavenging. The differences between wet and dry season convective processes are very clear; from the meteorological point of view, the dry season has higher lightning activity and warmer $T_{IR}$, and vice versa in the wet season.

To observe the diurnal modulation of PSD in more detail, the daily evolution and relative variation (variation around the mean value) were computed for $N_{UFP}$, $N_{AIT}$, and $N_{ACC}$ (Figure 5). $N_{ACC}$ and $N_{AIT}$ decrease during the night; the nocturnal BL height is normally above 60 m (Carneiro and Fisch, 2020), so this decrease is likely related to the interaction between the surface and the BL. Henkes et al. (2021) discuss the shallow and deep convective events. For both types of events, the nocturnal boundary layer has nearly zero and sometimes negative latent and sensible surface flux, as well as nearly zero turbulence kinetic energy, consequently no effective mechanism to support exchange between BL and free atmosphere. Therefore, this behavior can be attributed to dry deposition to the vegetation and ground surfaces, as there are no other sinks because the exchange across the top of the BL is very small due to the large stability in this layer. Fog, usual in the wet season early morning (Anber et al., 2015), could also contribute to this behavior, though the night-time decrease is a systematic behavior during the dry and wet season. Whereas $N_{ACC}$ and $N_{AIT}$ decrease during the night, the UFP number concentration remains nearly constant (dry season) or even increases in the wet season. This topic will be specifically discussed in the next section.

This behavior changes completely at sunrise, when accumulation and Aitken particles start to increase and $N_{UFP}$ decreases, with the typical behavior of a particle size growth process, from ultrafine to Aitken and eventually to accumulation mode sizes (note that $N_{AIT}$ starts to decrease at approximately 13:00 UTC (9:00 LST). As concentration also increases, there is also a contribution from the vertical transport to the BL. This process is observed up to the middle or end of the afternoon, when the concentration of accumulation particles begins to decrease and ultrafine particles increase in phase with the increase in the afternoon convective processes. Figure 5b shows the proportional diurnal variation, relative to the overall mean value, for the dry and wet seasons. It is interesting to note that despite their lower concentration relative to the accumulation mode particles, the UFP have the largest relative diurnal variation in both seasons. $N_{UFP}$ diurnally changes up to 40 % of the average number concentration. $N_{ACC}$, as already shown, varies in absolute number much more, but as the number concentration is very high, mainly in the dry season, the relative variation is very small. An exception occurs during the wet season, when $N_{ACC}$ is reduced by up to 30 % during the night. Thus, despite the much larger concentration in the dry season, the relative daily variability is larger during the wet season. This result indicates that the diurnal cycle not only mainly controls $N_{UFP}$, but also provides a modest decrease in the concentration of the larger modes during the night and shows a growth process from morning to the time that rainfall starts.

The diurnal cycle of $N_{UFP}$ (see Figure 3c) has an oscillation in phase among the two heights, with minimal values in the early afternoon. However, the diurnal cycle of $N_{ACC}$ (see Figure 3d) shows the number concentration at 60 m having its minimal value earlier than at 325 m, and a similar concentration in the early morning when the convective BL is developing. This feature shows the importance of the BL evolution in shaping the particle size distribution.

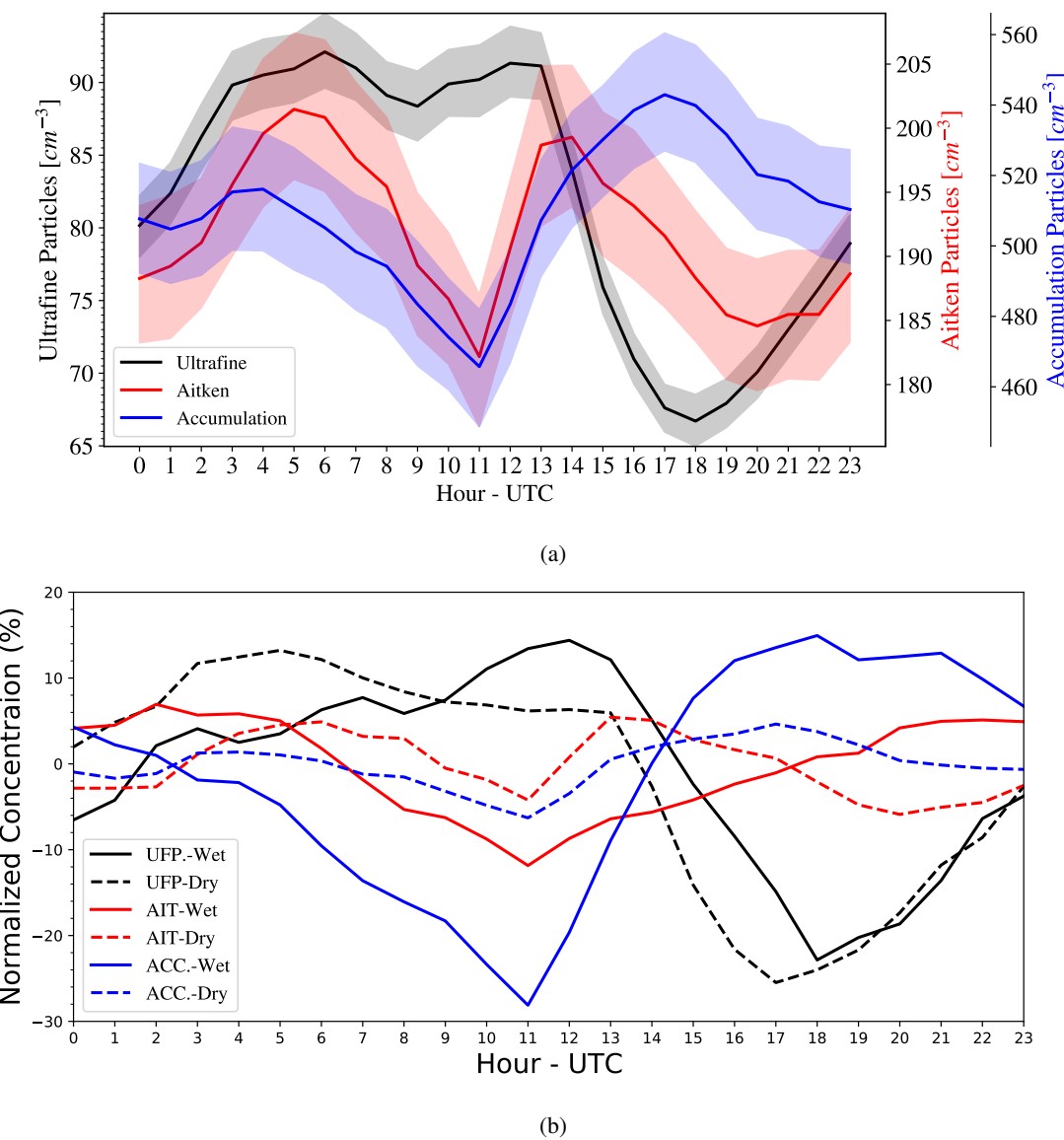

(a)

(b)

**Figure 5.** a) Diurnal cycle of the ultrafine (black), Aitken (red), and accumulation mode particles (blue) at 60 m sampling height. The shaded area corresponds to a confidence interval of 95 %. b) Mean deviation of the diurnal cycle of ultrafine, Aitken, and accumulation mode particles during dry (dashed line) and wet (solid line) seasons. Time is UTC, LST is UTC-4 hours.

### 3.3 Lightning Density and Particle Size Distributions

This section explores the relationship between lightning events and PSD. To understand how the PSD changes as function of the weather events, we evaluate the relationship between thunderstorms (i.e., storms producing lightning) and the evolution of $N_{\mathrm{UFP}}$, $N_{\mathrm{AIT}}$ and $N_{\mathrm{ACC}}$ from 400 minutes before to 400 minutes after the maximum lightning activity.

To describe the evolution of ultrafine, Aitken, and accumulation mode particles, composite analyses were produced based on the maximum lightning density activity. Composite analyses are useful tools to form physical hypotheses on the associations between environmental variables that occur over time (Boschat et al., 2016). Composite analysis is a useful technique to determine basic characteristics of a particular phenomenon. This type of analysis involves collecting large numbers of standardized cases of a given phenomenon and compositing them together as a collection. It involves computing the composite mean and statistical significance of the selected pattern, in this case the lightning occurrence and other cloud characteristics.

The composite depends on the specified threshold to create a collection of cases. For lightning, the threshold was computed as percentiles of the lightning density distribution. This procedure allows comparing different thunderstorm intensities and their effect on PSD, as well as the effect of the other parameters describing cloud activity, such as cloud top height, rainfall, and cloud liquid water. In order to have a comparable selection of lightning intensity and cloud properties, the lightning density selected was defined as a function of the percentile values of 25%, 45%, 65%, 85% and 95%; these percentiles correspond to lightning density events of 2, 30, 137, 494 and 1192 (number of lightning events in 10 minutes in an area of 5 by 5 pixels centered at ATTO), respectively. The composites were produced by selecting the particle number concentration 400 minutes before and after the maximum lightning density for the different values of the selected percentiles. Figure 6 shows the typical evolution of $N_{\mathrm{UFP}}$, $N_{\mathrm{AIT}}$, and $N_{\mathrm{ACC}}$ as a function of the maximum lightning density for different percentiles. Figure 6b also shows the composite of the lightning density for the different percentiles. The number of cases in each composite depends on the percentile and ranges from 2520 for 25% to 159 cases for 95%. Statistical significance was evaluated using t-student test and there is a significant difference (>99 %) between the different modes and percentiles.

At approximately 100 minutes before the maximum lightning density, for any lightning intensity, $N_{\mathrm{UFP}}$ particles increase in concentration, reaching maximum values of nearly double the initial concentration 200 minutes later. The difference in $N_{\mathrm{UFP}}$ increases exponentially with the lightning density percentiles, i.e., the thunderstorm intensity controls the rate of change of the ultrafine number concentration. Deierling and Petersen (2008) found a high correlation between volume updraft and total lightning. Jadhav et al. (1996) showed an increase in ozone ($O_3$) and nitrogen dioxide ($NO_2$) column densities during thunderstorm events. Oxidation products of volatile organic compounds form Secondary Organic Aerosol (SOA) to create new particles. Martin et al. (2010) discuss the mechanisms in the production of SOA in Amazonian forests and highlight the importance of nitrogen oxides and ozone in the oxidation processes and new particle formation. The increase in volume updraft also increases air mass flux exchange in mesoscale and consequently more frequent updrafts and downdrafts. Subsection 3.7 discusses these successive pulses of vertical movements forced by gravity waves. This cloud variability increases the frequency of downward transport, and consequently, the advection of ultrafine particles as suggested by Wang et al. (2016) and Andreae et al. (2018). In contrast, $N_{\mathrm{AIT}}$ and $N_{\mathrm{ACC}}$ decrease, reaching their minimum at nearly the same time as the maximum lightning

activity. The main reason for this change could be an exchange of PBL air with air from the free troposphere, which is higher in $N_{\text{UFP}}$ and depleted in $N_{\text{ACC}}$, or it could be due to the scavenging by the intense rainfall (more efficient for larger particles). These processes will be discussed in section 4.

The reason for this increase in concentration of the UFP cannot be answered with the experimental design of this study. Still we can hypothesize that the reason is related to the UFP being brought down by the convective cloud downdrafts. The layer of large concentration of UFP in the upper troposphere is above 10 km, but UFP, in lower concentration, is also found below this layer, where is more reasonable to find downdrafts in convective clouds. Considering the time interval between the onset of the lightning activity and the maximum UFP at the surface (60 m) is around 100 minutes and the UFP is advected from 8 km to the surface, the mean downdraft velocity would be 1.3 m s$^{-1}$. Schiro and Neelin (2018) show mean vertical velocity profiles for organized and isolated convective cells in the Central Amazonian region, with the downdraft velocities from the surface up to 10 km being between 1 and 2 m s$^{-1}$. This estimation could agrees with the results from Wang et al. (2016) and Andreae et al. (2018), suggesting that the source of these UFP particles observed at the surface is in the upper troposphere. However, as already mentioned, downdraft is mainly concentrated in the middle troposphere, we will show in section 4, that the Amazonian downdrafts are well below this upper tropospheric (10-14 km) layer, source of ultrafine particles. A further possibility that needs to be considered is the production of UFP by the lightning discharges themselves, as has been demonstrated for a site in China by (Wang et al., 2021)

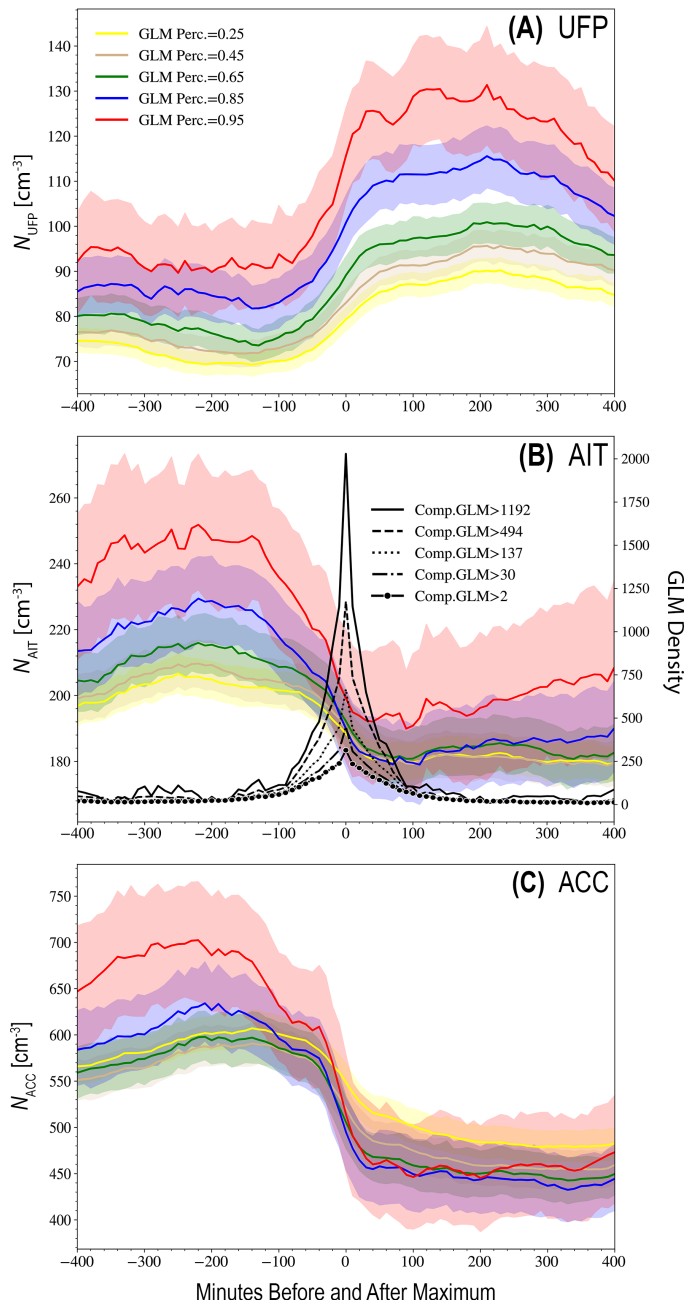

**Figure 6.** a) Composite $N_{\mathrm{UFP}}$ as a function of the maximum lightning density events for different lightning density percentiles. b) Composite $N_{\mathrm{AIT}}$ as function of the maximum lightning density events for different lightning density percentiles. The composite lightning frequency is shown for each percentile from continuous to dashed lines c) Composite $N_{\mathrm{ACC}}$ as function of the maximum lightning density events for different lightning density percentiles (60 m sampling height). The shaded area corresponds to confidence intervals of 95%.

### 3.4 Relationship Between Lightning Activity and Aerosol Background

As already mentioned, the changes in $N_{\mathrm{AIT}}$ and $N_{\mathrm{ACC}}$ happen in an opposite direction to $N_{\mathrm{UFP}}$; these particles are removed, and specifically, $N_{\mathrm{ACC}}$ also starts to decrease 100 minutes before the maximum of lightning density (the same time when $N_{\mathrm{UFP}}$ starts to increase). This observation indicates that the rainfall begins at approximately this time and intensifies up to the time of maximum lightning density, 100 minutes later. Interestingly, Figure 6 shows that even before the onset of scavenging (i.e., more than 100 minutes prior to the lightning density peak), a higher density of lightning is associated with a higher accumulation number concentration. Higher lightning activity appears to be associated with a higher aerosol background condition. The aerosol concentration earlier than 100 minutes before the maximum lightning event is $30\,\%$ higher at the 95th percentile value than that at the 25th percentile. Nonetheless, when approaching the time of maximum lightning activity and even later, the differences in $N_{\mathrm{ACC}}$ at the different lightning intensities become equal.

To understand this relationship of higher lightning activity at the time of higher aerosol background, a box-whisker diagram was computed relating lightning density and $N_{\mathrm{ACC}}$, 150 minutes before the maximum lightning event (Fig. 7). This time interval of 150 minutes before the intense weather event is an average time to detect the aerosol concentration before it is modified by the rain event. This figure highlights the positive correlation between the concentration of accumulation mode particles several hours before the maximum lightning activity and the increase in the frequency of the lightning events. During these periods of high activity convective events, with intense turbulence and well developed convective BL, the planetary BL is thoroughly mixed. The time interval considered could be related to the time needed to clean up all the atmosphere by rainfall. This is behavior could also explain the time interval between the beginning of the increase of ultrafine particles and the maximum lightning activity.

The relationship between high aerosol background and lightning activity is not clearly observed for all months of the year, although a trend of higher intensity lightning events related to high $N_{\mathrm{ACC}}$ is easily observed. Applying a t-student statistical test we can reject the null hypothesis for specific classes, indicating that there is a significant difference ($>95\,\%$) between the means of the classes with lightning density smaller and larger than 400 lightning events. During the period from April to August, this relationship is not clearly observed. However, when we look at the less polluted months from December to February (Fig. 7b), this trend is clearer but less significant because the number of lightning is smaller during this season. The reason cannot be answered with the present dataset, but the effect is better represented in this period than in a highly polluted situation. The sensitivity for invigoration should be higher at low aerosol loadings and become saturated at higher loadings (Rosenfeld et al., 2008). During May and April, the cleanest months (Pöhlker et al., 2018), this relationship is also observed but less pronounced than for the period from December to February. One possible explanation is the very low aerosol particle concentration that reduces the total number of CCN.

Lightning acting to help increase the concentration of UFP, which hours later grow to accumulation particles and in turn favor more lightning activity is likely related to the convective invigoration processes (Rosenfeld et al. (2008)). This is an interesting behavior, but as we will show, convection is mainly controlled by convective activity, which is strongly related to gravity waves and not to aerosol loading.

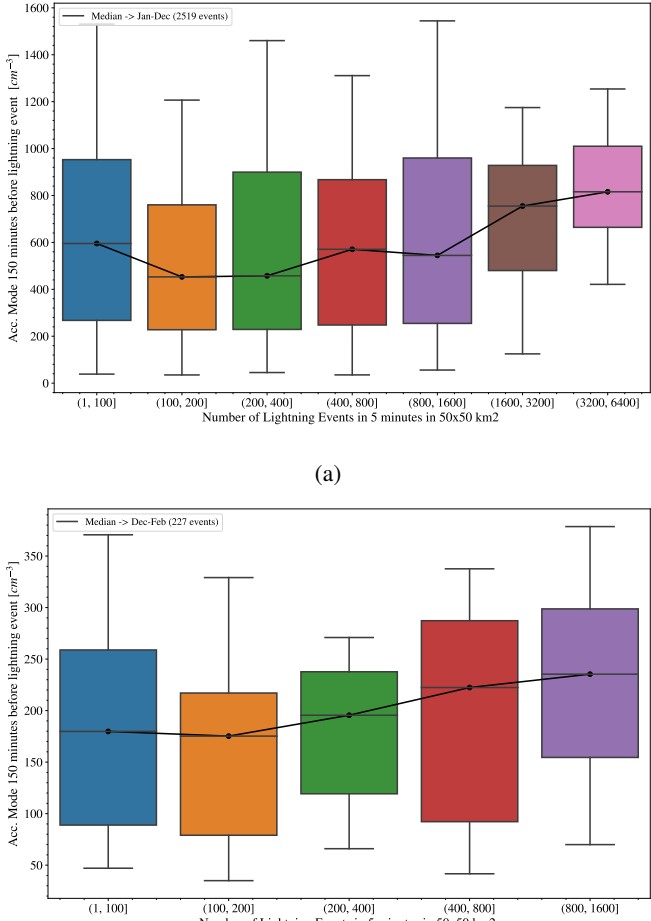

(a)

(b)

**Figure 7.** a) Whisker diagram between lightning density classes and accumulation particle mode number concentration (60 m sampling height), 150 minutes before the maximum lightning density for the two analyzed years. b) The same as (a) but for December to February.

## 3.5 Cloud Characteristics and Particle Size Distributions

The same kind of composite figures were prepared to analyze the effect of the rain rate (reflectivity at 3 km), cloud top height, cloud liquid water content, and brightness temperature on the PSD. In an effort to combine the different composites in the same figure, the deviations of the means were computed because the absolute magnitudes are different between these weather parameters. The composites are presented only for the maximum event defined by the threshold percentile value of 65 %. Different from the other variables, brightness temperature is a continuous value from clear sky to deep convective clouds. Therefore, a selection of the percentile value of 65 % would be associated with many thousands of values, in contrast to rainfall/lightning events, where the data are only reported for rainfall or thunderstorm (lightning case) events. Therefore the percentiles for $\mathrm{T}_{IR}$

were defined only for values smaller than 245 K, normally classified as the clouds associated with convective events (Machado and Rossow, 1993).

Figure 8 presents the evolution of $N_{\mathrm{UFP}}$, $N_{\mathrm{AIT}}$, and $N_{\mathrm{ACC}}$ from 400 minutes before to 400 minutes after the maximum of $T_{IR}$, vertically integrated liquid water (VIL), the 20 dBZ cloud top height (echo top), reflectivity at 3 km (rainfall), and lightning density, respectively, for wet and dry seasons. The results were split into dry and wet seasons to highlight the different behaviors. The effect on the evolution of particles is similar to the scenario shown for lightning, with a similar phase but with different amplitudes. The larger the amplitude, the more important is the effect of a specific meteorological variable in the modification of aerosol concentration. During the wet season, one can note changes around the mean value on the order of $25\,cm^{-3}$ for the number concentrations from all variables, with the exception of lightning density, which produces larger variations of approximately $40\,cm^{-3}$ in $N_{\mathrm{UFP}}$, $N_{\mathrm{AIT}}$ and $N_{\mathrm{ACC}}$. Thunderstorms that produce lightning are more intense than ordinary storms, and the stronger convective updrafts and downdrafts could explain this behavior.

The same feature is not observed for the dry season; lightning events are tied to similar or even smaller particle number concentration changes than the other variables. There are some possible reasons for this behavior, such as the large area of 5 by 5 pixels where lightning density is observed and the more localized convective cells during the dry season (Machado et al., 2018). This effect could likely minimize the variation of the number concentration at ATTO for smaller systems in the searched region that are farther from ATTO. One interesting point is the similar effect exhibited by the cloud liquid water, cloud top, and rainfall on the evolution of the particle number concentration. These variables are well correlated, but the general effect on particles appears to be mainly related to the rainfall processes and associated updrafts and downdrafts. Recall that GLM captures only lightning that is more intense and closer to the cloud top; therefore, the population of rainfall clouds could also be associated with less intense and shallower lightning events. A second aspect observed in this figure are the different amplitudes of the rainfall effect on the particle number concentration between the wet and dry seasons. For $N_{\mathrm{ACC}}$, for instance, echo top changes imply values of approximately $47\,cm^{-3}$ in the wet season versus $156\,cm^{-3}$ in the dry season.

We concluded that convective events have a typical behavior, increasing $N_{\mathrm{UFP}}$, and reducing $N_{\mathrm{AIT}}$, and $N_{\mathrm{ACC}}$. However, this PSD modulation by weather events depends on the intensity of the convective event and has different sensitivities during the dry and wet seasons.

### 3.6 The Rate of Change of PSD at 60 and 325 m

The preceding sections discussed how the concentrations of different particle modes evolve during convective events and how convective events are modulated by particle size concentrations. One way to evaluate in more detail how weather events modify the PSD is looking at the rate of change of the number concentrations of a specific mode as a function of lightning, at two different heights of 60 and 325m. The rate of change in the particle number concentrations ($N_{\mathrm{a}}$) at two different heights gives an idea of how the flux of particles is being modified at a given instant during the occurrence of a lightning event. For the purpose of evaluating this rate of change of a specific mode in response to weather events, we select the composite of the lightning density and compute the rates of change of the number concentrations. This rate is computed as $\frac{\mathrm{d}Na}{\mathrm{d}t}$ in units of $cm^{-3}\,hour^{-1}$. Figure 9 presents the rate of change of particles in the wet and dry seasons for the lightning density composite.

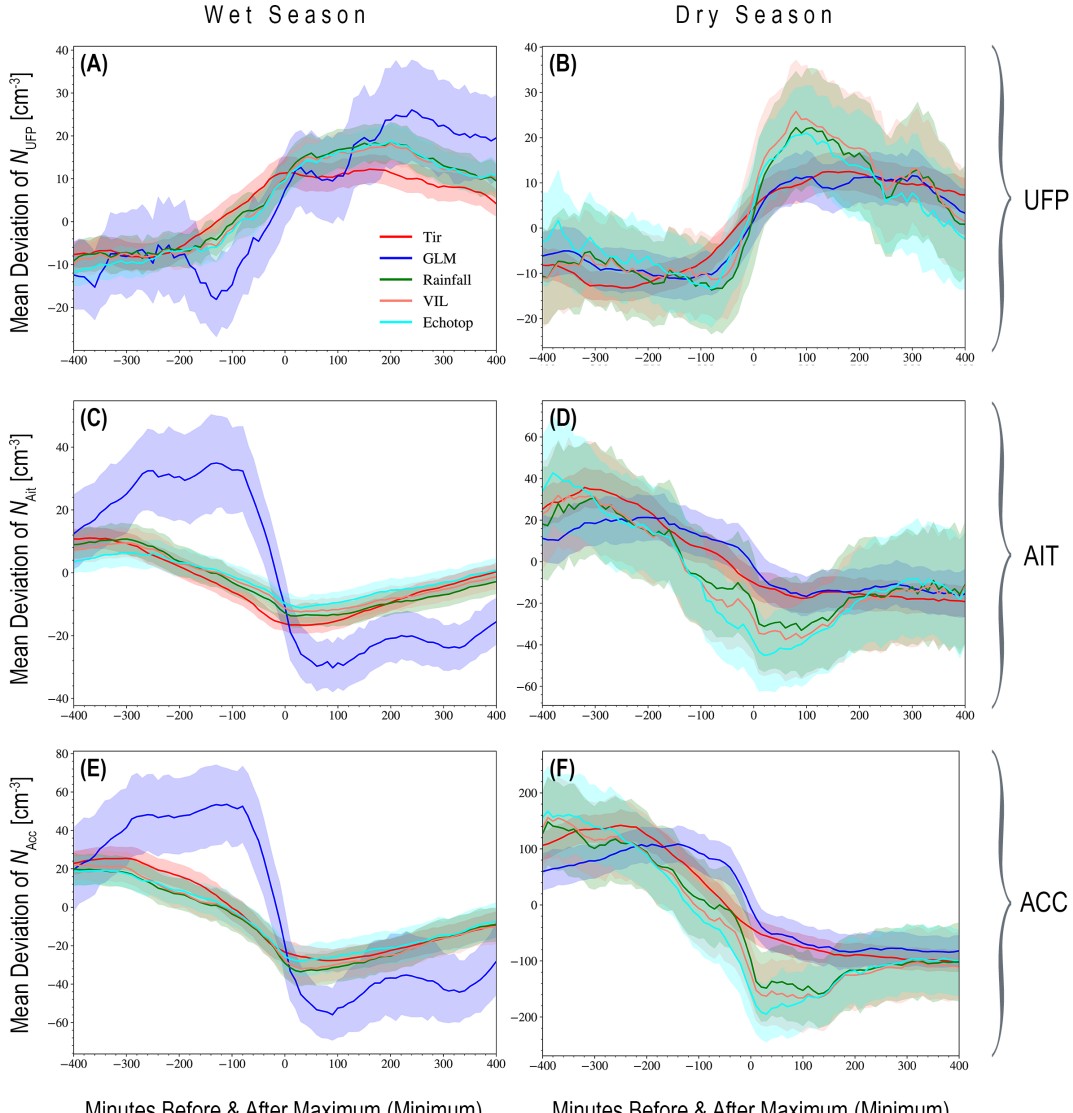

**Figure 8.** Composite reflectivity, vertically integrated liquid water (VIL), 20 dBZ echo top, brightness temperature, and lightning density for the percentile value of 65 % for a) $N_{UFP}$ during the wet season, b) $N_{UFP}$ during the dry season, c) $N_{AIT}$ for the wet season, d) $N_{AIT}$ for the dry season, e) $N_{ACC}$ for the wet season, and f) $N_{ACC}$ for the dry season (60 m sampling height). The shaded area corresponds to the 95 % confidence interval.

Some noteworthy features can be observed in Figure 9. First, UFP concentration changes have nearly the same phase in the wet and dry seasons; they are mainly negative between -400 and -200 minutes and between +200 and +400 minutes, and positive between -200 and +200 minutes. The rate of change in the wet season is larger and more than twice that of the dry season value at the time of maximum lightning density (minute zero). The rate of change at 60 m is higher than that at 325 m.

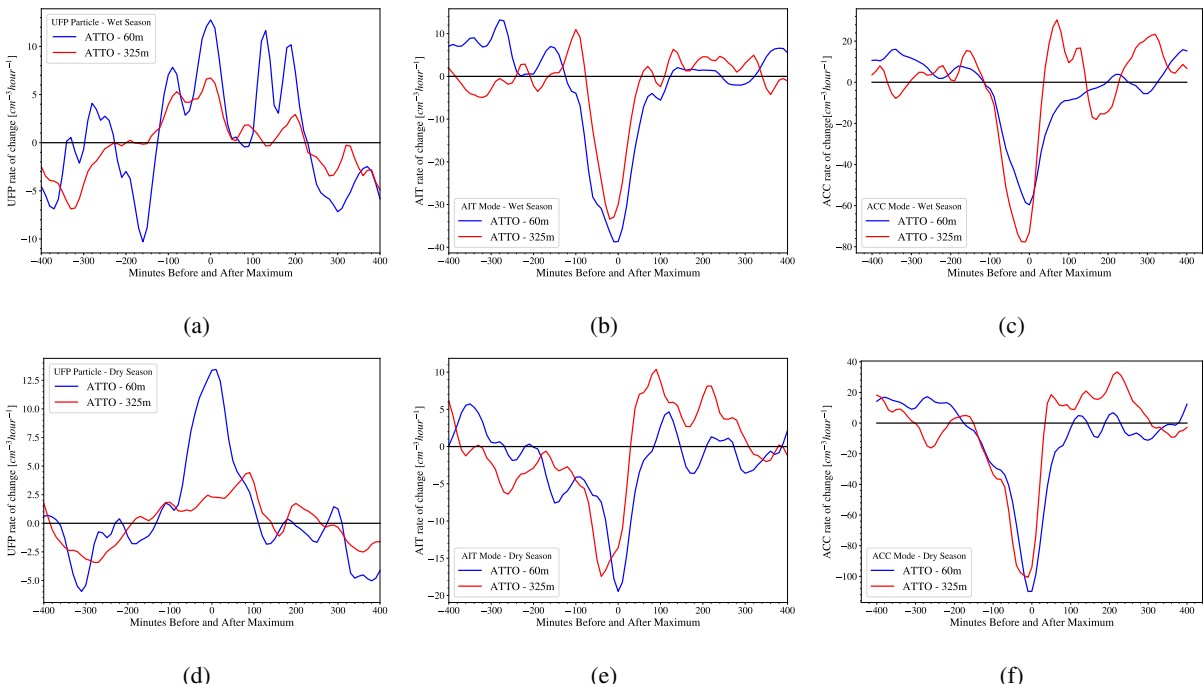

**Figure 9.** Rate of change in number concentrations from 400 minutes before to 400 minutes after the maximum lightning density (0.65 quartile) for a) $N_{\text{UFP}}$ at 60 and 325 m during the wet season, b) $N_{\text{AIT}}$ at 60 and 325 m during the wet season, c) $N_{\text{ACC}}$ at 60 and 325 m during the wet season, d) $N_{\text{UFP}}$ at 60 and 325 m during the dry season, e) $N_{\text{AIT}}$ at 60 and 325 m during the dry season, and f) $N_{\text{ACC}}$ at 60 and 325 m during the dry season

The ultrafine number concentration increases/decreases in one hour nearly 4 times more at 60 m than at 325 m. This observation indicates that the surface could also contribute as a source of UFP, increasing the UFP divergence flux in the BL. Even if the correction for the long inlet at 325 m is not sufficiently accurate to correct fully the losses of the concentration of small particles,

this derivative does not depend on the absolute value. It, therefore, confirms the different behavior between 60 and 325 m. For $N_{\text{AIT}}$ and $N_{\text{ACC}}$, in general, between -100 and +100 minutes there is a decrease in concentration and nearly no change during this period, except for $N_{\text{ACC}}$ later than +100 minutes, which shows a trend of increased concentration. For the accumulation mode particles, the rates of change of the number concentrations at 325 and 60 m have nearly the same magnitude. However, this effect is slightly larger during the dry season at 325 m and during the wet season at 60 m.

The rate of change also decreases with height. It is possible that when the convective BL is being built, UFP are advected to higher levels, or the dilution due to the growth of the well-mixed BL be the reason for such a decrease with height. When the sun rises, $N_{\text{UFP}}$ decreases, likely by growth into the Aitken mode particle range, reducing $N_{\text{UFP}}$ in the BL where the concentrations of the Aitken and accumulation mode particles are higher than that in the free troposphere (Schulz et al., 2018).

Another interesting feature is the high variability of the rate of change of particle concentrations. It is clear that the variability 395 is in the frequency of hours, indicating pulses of increase/decrease in the rate of change in the number concentration. This

intradiurnal oscillation observed in the rate of change of particle concentrations could be associated with the intradiurnal oscillation of the cloud cover. One of the main forcings of intradiurnal oscillation are the gravity waves. In the next section, we will explore the cloud modulation as well as the downdrafts and updrafts in this range of hundreds of minutes.

### 3.7  Cloud Intradiurnal Variability

The increase in UFP during convective events could result from new particle formation due to the increase of nitrogen oxides, ozone, and ions produced by the lightning discharges or by the vertical advection of UFP from upper levels. Both processes depend on the cloud dynamics, such as the downdraft/updraft intensity and frequency. Therefore, evaluating the cloud variability and what drives these oscillations is an important aspect of understanding how weather events modify PSD. Another aspect is the rate of change of the particle number concentration, in all modes, in response to weather events. It shows a well-marked
oscillation of order of hundreds of minutes with positive and negative rates at both heights. These successive pulses observed in Figure 9, here nominated as intradiurnal oscillation, could be the result of a series of updrafts and downdrafts or clear/cloud sky in a time interval of the order of hundreds of minutes. A specific analysis was performed to investigate this behavior. The main goal of this analysis is to verify if the cloud cover time-evolution has variability in this time range. In a region with smooth topography and homogeneous surface conditions, the forcing and trigger mechanisms of gravity waves produced by convective
towers could be the reason for this intradiurnal variability observed in the rate of change of the particle number concentrations.

Gravity waves modulate the deep convection, and at the same time, deep convection produces gravity waves that will modulate convection far away from the source. Lane and Zhang (2011) show that gravity waves control the variability of convective clouds. Vertical propagation of gravity waves produced by convective towers has an important effect on cloud variability (Alexander et al., 1995). In addition, strong updrafts induce tropopause fluctuations that are spread throughout the troposphere,
and latent heat release due to convection and interactions with the mean flow produce gravity waves (Grimsdell et al., 2010). Wang (2007) observed plume formation above thunderstorm anvils, and in simulating this case, they observed high instability and breaking of the gravity waves excited by the strong convection inside the storm. Using the day/night band from the Suomi National Polar-orbiting Partnership (NPP) satellite, Miller et al. (2015) shows several examples of thunderstorms triggering a broad spectrum of gravity waves.

Wavelet transforms were employed to evaluate if cloud cover variability has the same pattern as the rate of change of the particle number concentrations. The wavelet transform provides information about the time and the frequency, making it possible to evaluate the amplitude of a specific frequency range in time, and in this case, every 10 minutes. This methodology will create a time series of the amplitude of the frequencies in intervals with a range of 1-5 hours. The Morlet wavelet was applied to a frequency of $T_{IR} < 284\,K$, to account for the variability of the total cloud cover. The variability in 1-5 hours
was integrated and produced a 10-minute time series; the higher values of this time series were used as a proxy of high cloud oscillation in the interval of 1-5 hours. Like the composite studies employed in this study, we compute the maximum interdiurnal activity and perform composite studies.

Figure 10a shows the intradiurnal composite for the echo top and lightning density. We can see the high variability of the convection with a succession of moments of more intense convection, with a peak of lightning followed by a higher cloud top.

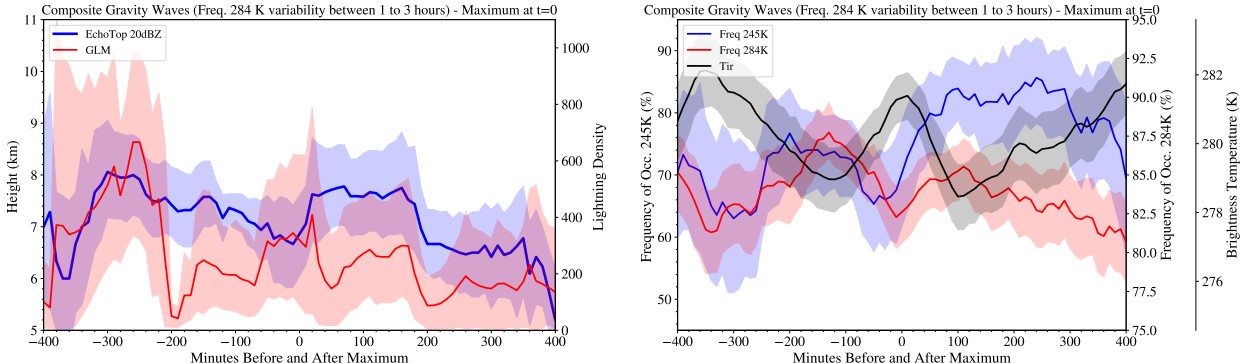

**Figure 10.** Composite at 400 minutes before and after the maximum of variability in the frequency interval from 1 to 5 hours of the total cloud cover ($T_{IR} < 284$K). a) Echo top and lightning density. b) $T_{IR}$, frequency of occurrence of $T_{IR} < 245$ K and $T_{IR} < 284$K in the area of 12 by 12 $km^2$. The shaded area corresponds to a confidence interval of 95%.

Figure 10b shows the composite for the parameters describing the cloud variability, $T_{IR}$, and the frequencies of occurrence $T_{IR} < 245$ K, a proxy for convective cloud cover, and $T_{IR} < 284$ K, a proxy for total cloud cover. This figure describes a typical variability in this frequency range. Convection in Amazonia, forced by this time scale, control the intracloud or intercloud variability and consequently the addition or loss of particles. During thunderstorms, the updrafts advect particles upward, followed by downdrafts advecting cloud content to the surface. This implies particle modulation by in-cloud scavenging of BL particles and by the downdraft bringing air rich in oxidants and/or UFP to the BL. This intermittent series of updrafts/downdrafts modifies the particles in the BL through a rich spectrum of processes, dominated by UFP enhancement and reduction of $N_{\text{AIT}}$ and $N_{\text{ACC}}$ that occur hundreds of minutes before and after the maximum convective events, not continuously but in a high variability environment.

## 4   Discussion

In the introduction, we referred to a large set of studies discussing aerosol-cloud-precipitation interactions. The present research brings new features to complement conceptual models of the effects of weather events on particle size distributions and vice-versa in the Amazonian region. However, several important questions are still open, and further studies are needed to develop a coherent conceptual model. This section will present some of the well-known processes, adding the new findings from the current investigation and putting in evidence other questions that are fully open.

One of the open questions is related to the source of UFP in the BL that nearly double in concentration during convective events. On one side, recent results show high concentrations of ultrafine particles in the upper troposphere (UT), although the formation processes are still unknown. Wang et al. (2016) described, for case studies, the phenomenon of sharply increasing $N_{\text{UFP}}$ and the simultaneous decrease in the $N_{\text{AIT}}$ and $N_{\text{ACC}}$ during rainfall events, in agreement with the composite studies presented in this study. They showed high concentrations of UFP in the FT, increasing with height, and proposed that convec-

tive downdrafts bring down UFP from the free troposphere into the BL. On the other hand, they suggest that these particles arise from new particle formation in the outflow regions of deep convective systems, followed by condensation and coagulation growth. Andreae et al. (2018) described the high concentrations of UFP in the UT based on the ACRIDICON-CHUVA campaign. They suggested that downdrafts transport these particles to grow into new CCN in the planetary BL by VOC oxidation and growth processes. However, downdrafts from deep convection are expected to occur preferentially below the 10-14 km,

layer of the source of high concentration of UFP. To evaluate this hypothesis, we analyzed the vertical velocity profile inside the clouds at the ATTO-Campina site. With the new radar wind profile, we have computed a statistical analysis of the vertical velocity. Figure 11 shows the contoured frequency by altitude diagram of the resultant vertical velocity for the thunderstorms season (October-November); therefore, it describes the period of maximum extension of the updraft vertical distribution. The instrument measures the vertical movement of the rain drops, which is a combination of the droplet terminal velocity and air

vertical motion. To provide a reference of the expected droplet terminal velocity, Martins et al. (2010) discuss rain droplets in the Amazonian region and mention peaks of 0.5, 1.0, and 2.0 mm size distributions for rainfall events, which approximately corresponds to a terminal velocity between 4 and 7 m s$^{-1}$. However, the main feature to be explored in this Figure is not the absolute value of the air vertical velocity, but the layer on which the vertical velocity acts. This Figure shows that downdrafts are mostly located below 10 km, but the layer of maximum concentration of UFP is mainly above 10 km. Wang et al. (2018)

computed the vertical velocity distribution for convective and stratiform clouds for the GoAmazon and confirm weaker composite vertical air velocity above 10 km. Therefore, the UFP are more prone to be advected through the Hadley and Walker cells to middle latitudes than to return to the surface tropics as discussed in Kida (1983) using a Lagrangian general circulation model of air parcels. However, the concentration of UFP in the upper troposphere is very high, order of magnitudes more elevated than the surface UFP. Therefore, even if the maximum downdraft is below the layer of maximum concentration, the

small fraction advected downward, bellow 10 km, could be enough to be the source of UFP particles at the surface during rainfall events.

   On the other side, the biogenic volatile organic compounds produced by forests undergo oxidation reactions to form secondary organic aerosol (SOA) further, and SOA accounts for a substantial fraction of tropospheric aerosol in the Amazon. Pfannerstill et al. (2021) show an increase in OH reactivity during rain events. The OH reactivity spikes have similar behavior

to the spikes of UFP during lightning events, and OH is an efficient mechanism to oxidize the biogenic volatile organic compounds and further form new particles in the BL. The mechanisms effectively describing the processes between rainfall events, the UFP in the upper troposphere, and the SOA formation close to the surface should be investigated to clarify these aspects.

   If the new particle formation from thunderstorm events presents these myriad of processes from the UFP in the upper troposphere to the VOC production by the forest, the daily cycle of new particle formation appears to be very clear. There

is a typical diurnal cycle behavior, $N_{\text{UFP}}$ exhibits a maximum concentration during the night; and as soon as the sun rises and the convective BL is built, conversion processes to Aitken and accumulation begin, and particle growth occurs up to the time that convection is initiated. Franco et al. (2021) studied ultrafine particle growth and found the most frequent starting cases approximately at twilight time before the convective BL is developed. Consequently, there is likely a combination of effects, such as the photochemical effect that begins the processes of UFP growing into the Aitken particle size range and the

development of the convective BL that contributes to the dynamics of the aerosol (upward advection). In the late afternoon, larger particles start to decrease by rain out, scavenging, and updraft removal. This process of increasing UFP and decreasing Aitken and accumulation particles occurs over a large range of variability with successive rain cells, updrafts, and downdrafts, controlled by the gravity waves. The rainfall scavenges the larger particles and the associated downdraft contributes to the increase in $N_{\mathrm{UFP}}$.

Related to the seasonal variation, one can note that in the wet season the weather-PSD interaction is stronger than in the dry season. For instance, for the diurnal cycle, the percentage of relative variability of $N_{\mathrm{UFP}}$ is similar, but the consequent effect on $N_{\mathrm{ACC}}$ is only observed during the wet season. It could be associated with the reduction of concentration during the night followed by a growth process from UFP to accumulation mode particles, in the early morning, that is around 30 % in the wet season and nearly imperceptible in the dry season due to the very high absolute concentration of accumulation

mode particles in the dry season. Also, related to the effect of lightning on PSD, the effect during the dry season is small and shows a small difference among the different types of rainfall events (storm, thunderstorm, cloud top height, total liquid water). The dry season is the maximum lightning activity season, but it is in the wet season that a relationship between lightning and $N_{\mathrm{ACC}}$ (150 minutes before the maximum lightning event) is observed. Hernández Pardo et al. (2021) using observation and modeling for Amazonian clouds, concluded that the width of the cloud droplet size distribution for clean

clouds (aerosol concentration $< 900\,\mathrm{cm}^{-3}$), varies as function of the adiabaticity of the cloud. However, for polluted clouds (aerosol concentration $> 2000\,\mathrm{cm}^{-3}$), the distribution width is not a function of the adiabaticity, and varies very little. There are clear differences between the seasons as shown in the present study, the high concentration of particles during the dry season is relatively less sensitive to weather conditions than during the wet season, with lower aerosol concentrations, when the different types of weather events dominate the weather-PSD interactions.

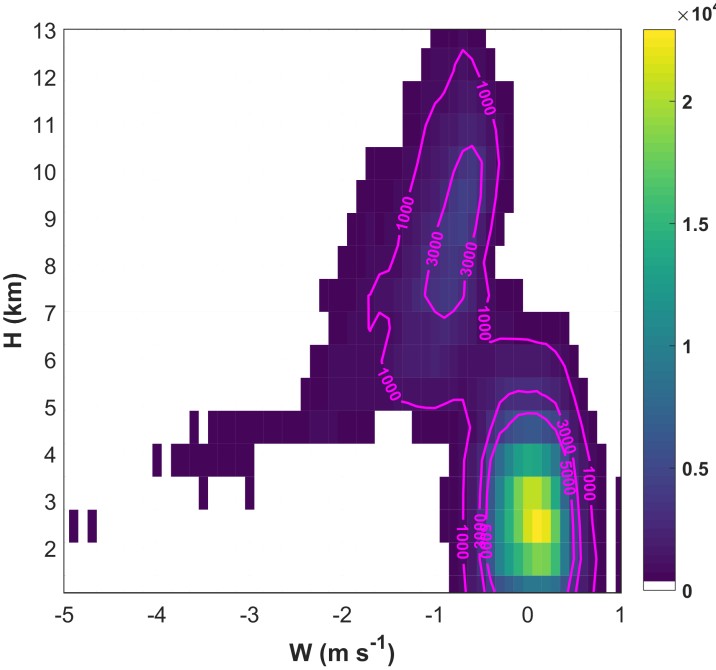

**Figure 11.** Contoured frequency by altitude diagram of the vertical velocity $W$ obtained by the RWP collected during 54 days of measurements between October 16 and December 8, 2020. The vertical resolution is 840 m, while the $W$ resolution is $0.1\,ms^{-1}$. The contours show the number of data points (magenta curves).

## 5 Conclusions

Particle size distribution data from two SMPS instruments at ATTO at 60 and 325 m was combined with satellite and radar data to provide the basis for exploring the relationship between weather events and changes in particle size distributions in the BL of the Central Amazon forest. A colocated time series for ATTO at 10-minute resolution was produced, combining the different types of measurements. Several features describing the evolution of the particle size distribution could be observed for different weather cycles, forcings, and characteristics.

The seasonality is pronounced, with particle concentrations in the dry season being an order of magnitude greater than that during the wet season, and the brightness temperature followed this behavior, with colder brightness temperatures during the wet season (cold cloud tops) than during the dry season. However, lightning has a completely different behavior, and the transition season is the period where the maximum lightning activity is observed.

The diurnal cycle shows a typical behavior with nearly the same phase in both seasons. The maximum concentration of UFP is observed at sunrise, and at this time, $N_{\mathrm{AIT}}$ and $N_{\mathrm{ACC}}$ present a minimum. As the sun rises and the convective BL develops, the UFP begin to show a decrease in the number concentration followed by an increase in $N_{\mathrm{AIT}}$ and $N_{\mathrm{ACC}}$, with a typical behavior of growth in the particle size. At the end of the afternoon, an inverse situation occurs, and $N_{\mathrm{AIT}}$ and $N_{\mathrm{ACC}}$ decrease

while $N_{\text{UFP}}$ increases due to the more intense rainfall. Despite the higher concentration of accumulation particles, it is $N_{\text{UFP}}$ that has the larger relative amplitude, except during the wet season in which the concentration of accumulation particles mode also has a significant relative variation.

The composite with the lightning activity describes the typical characteristics of the evolution of particle size changes due to convective rainfall. At approximately 100 minutes before and 200 minutes after the maximum lightning activity, $N_{\text{UFP}}$ increases and Aitken and accumulation particles decrease. Another interesting property is the larger number of lightning events when the aerosol background is higher, which appears to be an indication of convective cloud invigoration.

The same behavior as for lightning is observed in the composites of the echo top, rain rate (reflectivity at 3 km), integrated cloud liquid water (VIL), and brightness temperature ($T_{IR}$). Notably, a different behavior is observed between wet and dry seasons. During the wet season, lightning correlates with a greater change in the particle size distribution in comparison with the dry season for the other weather parameters. This feature should be investigated in future campaigns to explain the reasons for this large sensitivity during the wet season. It seems that for small aerosol concentrations, weather-PSD interaction is dominated by the weather events, however, during the dry season this interaction is dominated by the total aerosol concentration.

The interdiurnal variability modulates the convective activity and produces a high variability at hourly time scales, making the particle size distribution evolution much more complex and controlling the succession of rain cells, updrafts, and downdrafts. The increase of UFP and reduction of Aitken and accumulation mode particles is modulated by a secession of updrafts/downdrafts and cloud types.

Finally, the vertical distribution of the vertical velocity was computed using the radar wind profiler at the ATTO-Campina site. The data were collected in the season of the most intense convective activity (October – December) and show that downdrafts are mainly located below 10 km, while the layer of maximum concentration of UFP is mainly above 10 km. However, if even a fraction of UFP from the middle levels of the troposphere is advected downward, it could support the concentration of UFP in the surface observed during convective events. $N_{\text{UFP}}$ in the upper troposphere is several orders of magnitude higher than at the surface, but in the middle level of the atmosphere, there is still have a concentration of UFP larger than in the surface. Even though, the source of the UFP at the surface during convective events is still an open question because there is also the mechanism of particle formation related to the increase of OH and NOx during convective events.

This study clarifies the processes related to modulation of PSD and new particle formation associated with the seasonal and diurnal cycle, intradiurnal variability, and different types and characteristics of convective events. But, it opens further questions that need to be pursued in detail in new field campaigns, such as CAFE-Brazil scheduled for 2022 and the new ATTO-Campina site that provides measurements of atmospheric and cloud physical properties.

*Data availability.* ATTO data can be found in the ATTO data portal under https://www.attodata.org/ (ATTO, 2020). For data requests beyond the available data, please refer to the corresponding authors.

*Author contributions.* LATM conceptualized this study, conducted the data analysis and wrote the paper. MAF, LAK, FD, and BAH conducted the SMPS measurements at ATTO and took care for data quality checks and processing. MLP, PA, and SW supported the ATTO measurements and data analysis. MAC contributed with the LAP3000 data treatment and analysis MOA, UP and CP contributed to the interpretation of the data and finalization of the paper in various discussions. All the authors contributed to the interpretation of the results and writing of the paper.

*Competing interests.* The authors declare that they have no conflict of interest.

*Acknowledgements.* This research has been supported by the Max Planck Society, the FAPESP grant 2017/17047-0, the Bundesministerium für Bildung und Forschung (BMBF contracts 01LB1001A and 01LK1602B), and the Max Planck Graduate Center with the Johannes Gutenberg University Mainz (MPGC). For the operation of the ATTO site, we acknowledge the support by the Max Planck Society, the German Federal Ministry of Education and Research and the Brazilian Ministério da Ciência, Tecnologia e Inovação as well as the Amazon State University (UEA), FAPEAM, LBA/INPA, and SDS/CEUC/RDS-Uatumã. MAF acknowledges CNPq PhD grants 169842/2017-7, and CAPES 88887.368025/2019-00. MAC was supported by FAPESP grant 2020/13273-9. We would like to especially thank all the people involved in the technical, logistical, and scientific support of the ATTO project, in particular Susan Trumbore, Carlos Alberto Quesada, Reiner Ditz, Jürgen Kesselmeier, Thomas Klimach, Björn Nillius, Antonio O. Manzi, Thomas Disper, Hermes Braga Xavier, Nagib Alberto de Castro Souza, Adir Vasconcelos Brandão, Amauri Rodriguês Perreira, André Luiz Matos, Fábio Jorge, Fernando Goncalves Morais, Roberta Pereira de Souza, Bruno Takeshi, Uwe Schultz, Karl Kübler, Olaf Kolle, Martin Hertel, Kerstin Hippler, and Steffen Schmidt. In particular, we would like to thank Delano Campos, Andrew Crozier, Sam Jones, Juarez Viegas, Wallace Rabelo Costa, and Antonio Huxley Melo Nascimento for the frequent maintenance and troubleshooting of the ATTO instruments. We would like to thanks the comments and recommendations of the two anonymous reviewers.

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
