# Peer review of "How weather events modify aerosol particle size distributions in the Amazon boundary layer"

_Atmospheric Chemistry and Physics, 2021_

## Referee Comment (RC1)

Review on
**"How weather events modify aerosol particle size distributions in the Amazon boundary layer"**
by
Luiz A. T. Machado et al.

General:

This manuscript describes observations of potential influence of the daily weather cycle on the occurrence of aerosol particle modes in the Amazon basin. The observations are based on the tall tower ATTO accompanied by radar and satellite data.

This is definitely an interesting data set to analyze. Unfortunately, I do not survey the current literature, so I cannot pass judgment on the novelty of this data set per se, but only on the underlying manuscript and the data analysis described therein. Unfortunately, I see serious problems with the current version of the manuscript, which I will specify in the following. My main criticism is about the general structure and the structure of the subsections.
I suggest giving at least a short overview in each paragraph and a motivation for the following paragraph before entering into a discussion where the reader does not know where exactly it should go. I also suggest that in the results section you really focus on your own observations and not start with a literature review - the latter belongs either in the discussion or even in the introduction.

As for the content, I have big concerns with the chapter about the influence of gravity waves. If gravity waves are controlled by deep convection, I don't understand why they are discussed again when deep convection itself and its effects on aerosol population have already been discussed in detail.

From my point of view this manuscript needs careful rewriting at many places (see specific comments) but I highly encourage the authors to resubmit a modified version. As I said, this dataset has a lot of potential and the effort of a revised version should be limited. Major and minor specific comments are listed below:

Specific Comments:

Line 72: I fond the descriptions of the SMPS somewhat chaotic; it is more like a numbering of type numbers and a read not familiar will not be able to understand your setup.

Around l 75: Why do you use different driers for the two inlet lines – is there any influence on the results?

Line 89ff: I do not understand what you mean here, can you explain with a little bit more detail? With the figure it becomes clearer to me but maybe one sentence would help to explain what you mean. Why not showing an averaged size distribution including mode fitting?

More general: I know from many other review processes that it becomes more and more popular to use abbreviations for everything. Of course this is a question of style but I feel that a text becomes less readable if everything is abbreviated. Are these abbreviations such as ACCP really necessary - I would avoid trying to abbreviate everything!? See also the table. I

suggest to abbreviate parameters such as $N_{ACC}$ but not to abbreviate everything in the continuous text such as ACCP or so.

L 94: Why not citing the original paper by Hoppel?

Sec 2.3 "GOES-16": one general sentence at the beginning describing the general sensor type would be very helpful. You jump directly into technical details and only experts will know that you are talking about a satellite-based system. Furthermore, I suggest not using the model type of an instrument as a subtitle

Please check the sentence in line 132:

Section 3 "Results and discussion" is a mixture between discussion of previous findings, discussion and presenting own observations which make is hard for the reader to follow and at least at a few points the reader is lost and does not know which type of results you discuss. I strongly recommend restructuring Sec 3

Line 150ff: If you start with a discussion on "total particle number concentration" why do you refer to a fig in the appendix of accumulation mode particles? Why do you cite you own co-authors in this context?

And if the figure is worth to cite, I suggest to including it in the main body text. I never understood why there is an appendix based on four figures without any explanation and the subtitles needs much more explanation.

Isn't it trivial that CCN is based on aerosol size distribution and number concentration, as CCN is a part of the aerosol population?

Figure 2 has different fonts and size, which are partly way to small; the given precision of diameters is too high

Line 158: I suggest: Tir => $T_{IR}$   ?!

Line 163 a temperature is „higher" but never „warmer", same sentence: why can be a relationship „straightforward" it can be clear or so...

L 165: „*However, the lightning activity is highest at the same time that the aerosols have the maximum concentration, indicating that the connection between them is more complex, as will be subsequently discussed in the paper.*" Why indicates a relationship that the connection is complex? I cannot follow this argumentation

L 168: again you refer in the text to a figure in the appendix – you are in the analysis section where I expect to see an analysis of your data – why is this data figure not in sec 3 ?

L 169: I think it would be very helpful to know the accuracy of the correction for sampling losses – then you can answer the question...

Line 185: this is somewhat repetitive and can be shortened as it is classical BL theory and you have the appropriate citations already in.

Line 194 although I have a feeling of what you mean you should clearly define what you mean with things like „relative variation"

Line 195 „nocturnal BL => residual layer?

Line 196ff: this argumentation is really hand waving: Do you have any information about the inversion strength and possible wind shear which determine how effective entrainment can be? Your description at this point is very speculative. Furthermore, the exchange of any scalar property strongly depends on the strength of the gradient of the scalar itself as $<w'N'>$ scales with $d<N>/dz$.

Line 198ff: *"Fog could also contribute to this behavior, though the night-time decrease is a systematic behavior during dry and wet season."* Do you have any information about fog events? Otherwise this is again pure speculation.

Fig 4; it is difficult to realize which axis label belongs to which parameter – probably the right panel is shifted to far left and mask the label of the left panels?

Fig 3: I am not sure if it is a convincing idea to compare the temporal evolutions of size distributions (wet & dry) with different methods (color code & isolines) – this is very irritating.

Line 236/7 figure numbers are missing

L 249 "cloud" => "could"

Line 240ff: is there any more information of vertical wind velocity at the tower available (ultrasonic anemometer?) – this could help to understand the situation in more detail – just an idea.

Line 243/4: The sentences do not really add to the discussion. Is it surprising that there is a correlation between updraft and lighting? - Probably not. And why is O3 and NO2 of any interest here? Does this help to interpret your observations? For me it looks like you want to argue that those gases trigger nucleation? Or are those gases considered as tracer for vertical motion? There is a clear red line in your interpretation missing although the observations are definitively interesting!

Maybe one idea would be to investigate one case in more detail to corroborate your hypothesis?! Concerning Fig 5 one would expect a rapid decrease of total number concentration in case of lighting events – right? Are there total counters with 1Hz-resolution available in 60 and 300 m at ATTO? I think it could be convincing to analyze such an event as a case study and then extrapolate the findings with a statistical analysis as you did.

Line 245: how can a volume updraft result in an increasing downward advection of ultrafine particles? This sentence does not make any sense to me – please verify.

General: GLM stands for a sensor/device – right? Sometimes you use this abbreviations being a parameter (GLM density)…please specify what you really mean and be consistent.

L 250ff: Assuming the source for high $N_{UFP}$ is in the upper troposphere in about 10 km or so and you have a maximum concentration observed at ground 100 min after the onset of deep

convection with lightning this would imply a mean "effective" vertical downdraft of about 1.6 m/s which seems to be realistic. I suggest a few more (rough) estimates like this to see if hypotheses are at least realistic or not.

Line 264: I cannot follow this argumentation, which is mainly due to the fact that Fig. 6 needs better explanation about what exactly you are doing here. And are you sure the phrase "feedback" is correct in this context?

Line 258 : reference is missing

What else do we learn more about possible processes from Fig 6 compared to Fig 5? I think the correlation between lightning and aerosol concentration is also visible in the graphs of Fig 5 (which I really like).

Line 268 I am not sure if we can really should call this a "lag": with the first observed lightning we see already an increase in particle concentration and - following your argumentation – strong mixing begins. If we have diluted the lower atmosphere it is not so important anymore if you increase lightning/mixing as the PBL is already thoroughly mixed/diluted. Maybe I am wrong but wouldn't this also explain why the maximum in lightning is shifted to the change in aerosol concentration?

By the way, why not distinguishing in Fig 5 between "dry" and "wet" season as in Fig 7?

Line 312: such information can be shifted into the figure caption

Line 318: "*This observation indicates that the surface could be the source of the divergence flux of new UFP*." What do you mean with this sentence? How can the surfaces being the source of a flux? Please clarify.

Line 325: Could dilution due to the growing well-mixed BL also be a reason?

Line 333: I suggest „deep convection **can** produce gravity waves" – I think there need some conditions to be fulfilled for gravity waves. Furthermore, what do you mean with "vertical gravity waves"? The propagation? Please specify.

Beginning of Sec 3.4: over seven lines you cite papers describing observations of gravity waves – but why? There is no motivation for analyzing gravity waves so far. Why are you interested in this phenomenon? This should be placed first!

Line 340: Why applying wavelet transformation? There are some basic questions missing at the beginning of this section.

Line 341: "…to a frequency of $T_{ir} < 284$ K,..." – this is misleading, frequencies usually don't have the unit Kelvin – please rewrite this sentence

Line 349: "..*Intracloud or intercloud variability controls convection in Amazonia and consequently the formation or reduction of particles*." I would argue that convection controls the cloud variability and not the other way around. By the way, particle formation is something different compared to increased concentration due to (vertical) transport. Did you discuss particle formation, which is usually a nucleation process? And what has this to do with gravity waves, which are the subject of this paragraph? Sorry to say but this part is a mess and needs to be completely re-written.

Sec 3.5 starts again with an incoherent list of quotes regarding nucleation without telling the reader what this section will be about. Instead in line 365 a figure is introduced about vertical velocity observations – where is the red line in this section?

Line 365: „contoured frequency by altitude diagram" really does not need a shortcut.

The following part includes a lot of technical details that could be shifted to the technical part at the beginning of the manuscript (Sec 2.4)

Line 370ff: Now I am confused: what exactly is the „vertical velocity w" shown in Fig 10: The velocity of raindrops or the wind velocity?   The distribution is significantly skewed to negative velocities so I assume it is the drop velocity? I need to understand first what is displayed before making any comments on the scientific interpretation.

Line 3745: What do you mean by this sentence? If Fig. 10 shows the droplet velocities, no statement can be made about the cloud dynamics, or am I fundamentally misunderstanding something?

Line 376: Which observations tell you that the maximum $N_{UFP}$ is above 10 km in your observational period?
This paragraph does not provide a conceptional model based on your observations but rather on speculations partly based on other publications with results not necessarily valid for your situation.

I don't make any detailed comments about the conclusion section as the manuscript needs major re-writing and it is not clear at this state in which direction the data interpretation will finally go.

---

## Author Comment (AC1)

**Response to Anonymous Referees**

**Article: "How weather events modify aerosol particle size distributions in the Amazon boundary layer", by Luiz A. T. Machado et al.- ACPD 2021314.**

Dear Editor,

The authors would like to thank both reviewers for their helpful comments and suggestions. Our point-to-point responses are developed hereafter, along with an indication of changes made in the revised version of the text.

. As a summary, the revisions to the manuscript include the following highlights:

- We have clarified the text. We modified the text according to the recommendations, explaining the scientific aspects and introducing each section. We also added two more subsections and renamed the former Conceptual Section to the Discussion Section. This new section was reorganized to clarify the aspects we know, the new results from this study, and the open questions.
- The gravity waves are better introduced in the text now. Thanks for the comments; the text clearly explains why gravity waves are included in the manuscript in this new version. Particle size distribution is evaluated with the diurnal, seasonal, and intradiurnal cycles. We show that the cloud intradiurnal oscillation has the same frequency oscillation as the rate of change of the particle number concentration. Further, we discuss that this intradiurnal oscillation is forced in response to the gravity waves.
- T-student tests were applied to composite studies.
- Both reviewers recommend we split the former figure 3 into two figures, one for each season. Therefore, we added in this letter the two suggested figures and the former one and try to convince the Reviewers that the actual figure brings the opportunity to compare both seasons from the particles size distribution, lighting density, and brightness temperature. However, I am fully open to follow the suggestion to change to one figure for each season. Therefore, in this new version, we maintained the figure is the former version.
- Supplementary figures were incorporated into the text.

The individual reviewer comments and responses are included in the following document (author comments in *italics*, reviewer comments in **bold**).

Sincerely,

Luiz Machado, on behalf of all co-authors

**Reviewer 1:**

**Main comments:**

**General:**

**This manuscript describes observations of potential influence of the daily weather cycle on the occurrence of aerosol particle modes in the Amazon basin. The observations are based on the tall tower ATTO accompanied by radar and satellite data.**

**This is definitely an interesting data set to analyze. Unfortunately, I do not survey the current literature, so I cannot pass judgment on the novelty of this data set per se, but only on the underlying manuscript and the data analysis described therein. Unfortunately, I see serious problems with the current version of the manuscript, which I will specify in the following. My main criticism is about the general structure and the structure of the subsections. I suggest giving at least a short overview in each paragraph and a motivation for the following paragraph before entering into a discussion where the reader does not know where exactly it should go. I also suggest that in the results section you really focus on your own observations and not start with a literature review - the latter belongs either in the discussion or even in the introduction.**

**As for the content, I have big concerns with the chapter about the influence of gravity waves. If gravity waves are controlled by deep convection, I don't understand why they are discussed again when deep convection itself and its effects on aerosol population have already been discussed in detail.**

**From my point of view this manuscript needs careful rewriting at many places (see specific comments) but I highly encourage the authors to resubmit a modified version. As I said, this dataset has a lot of potential and the effort of a revised version should be limited. Major and minor specific comments are listed below:**

*Dear Reviewer, we would like to thank your suggestions and comments, they were essential for improving and clarifying some critical aspects of the manuscript content. In the Editors letter, we explain the main changes in the manuscript, and below, we listed these aspects related to your recommendations.*

*We have attempted to clarify the text. We modified the text according to the recommendations, describing the scientific aspects and introducing each session. We also added two more subsections and renamed the former conceptual section to Discussion Section. This new section was reorganized to clarify the aspects we know, the new results from this study, and the open questions.*

*The Gravity waves were better introduced in the text.* Thanks for the comments; the text clearly explains why gravity waves are included in the manuscript in this new version. *Particle size distribution is evaluated with the diurnal, seasonal, and intradiurnal cycles. We show that the cloud intradiurnal oscillation has the same frequency oscillation as the rate of*

*change of the particle number concentration. We discuss that this intradiurnal oscillation is forced in response to the gravity waves.*

**Specific Comments:**

**Line 72: I fond the descriptions of the SMPS somewhat chaotic; it is more like a numbering of type numbers and a read not familiar will not be able to understand your setup.**

*We have clarified this section. In essence, we describe here a standard SMPS system that has been running over long periods at ATTO. Since the instruments fail rather frequently in this challenging environment, the components of the SMPS setup (i.e. DMA and CPC) have been replaced several times. In the course of these replacements, different DMA and CPC models were applied. Note that these changes in the setup had no influence on the data comparability according to our frequent checks and cross-comparisons.*

**Around l 75: Why do you use different driers for the two inlet lines – is there any influence on the results?**

*Since 2021, we have tested and used different types of automatic dryers for the sample air under harsh ATTO conditions. The dryers mentioned in the manuscript are those that have performed exceptionally well over long periods. Furthermore, we have no indications that the different driers influence the data (e.g., through enhanced particle losses).*

**Line 89ff: I do not understand what you mean here, can you explain with a little bit more detail? With the figure it becomes clearer to me but maybe one sentence would help to explain what you mean. Why not showing an averaged size distribution including mode fitting?**

*The Reviewer refers to the following statement: In order to evaluate the particle characteristics along with the classes, we computed the cross-correlation between the different particles sizes for the Central Amazonian dry (Jun to Oct) and wet (Dec to Apr) seasons. Figure 1 shows how well the aforementioned modes describe the multimodal PSDs by means of cross-correlation plots. The cross-correlations show how the modes are correlated among the different particle sizes, clearly separating the distinct modes (that are not correlated). The plot for each season is to show how these characteristics change between the seasons.*

**More general: I know from many other review processes that it becomes more and more popular to use abbreviations for everything. Of course this is a question of style but I feel that a text becomes less readable if everything is abbreviated. Are these abbreviations such as ACCP really necessary - I would avoid trying to abbreviate everything!? See also the table. I suggest to abbreviate parameters such as NACC but not to abbreviate everything in the continuous text such as ACCP or so.**

*That is correct – abbreviations could help to shorten the text but should not be used excessively. In this case, we changed the acronym of Aitken and Accumulation particles modes as suggested.*

**L 94: Why not citing the original paper by Hoppel?**

*It was changed as recommended.*

**Sec 2.3 "GOES-16": one general sentence at the beginning describing the general sensor type would be very helpful. You jump directly into technical details and only experts will know that you are talking about a satellite-based system. Furthermore, I suggest not using the model type of an instrument as a subtitle**

*It was changed as recommended.*

**Please check the sentence in line 132:**

*The text was changed.*

**Section 3 "Results and discussion" is a mixture between discussion of previous findings, discussion and presenting own observations which make is hard for the reader to follow and at least at a few points the reader is lost and does not know which type of results you discuss. I strongly recommend restructuring Sec 3**

*This section was subdivided into more subsections, and an introduction to each section was added, and the Discussion Section was clarified.*

**Line 150ff: If you start with a discussion on "total particle number concentration" why do you refer to a fig in the appendix of accumulation mode particles? Why do you cite you own coauthors in this context?**

*Figures were added to the text, and the sentence was clarified.*

**And if the figure is worth to cite, I suggest to including it in the main body text. I never understood why there is an appendix based on four figures without any explanation and the subtitles needs much more explanation.**

*Changed as recommended, figures were included in the text.*

**Isn't it trivial that CCN is based on aerosol size distribution and number concentration, as CCN is a part of the aerosol population?**

*The CCN can also be a function of the hygroscopicity and chemical composition, but it matters only during a few episodes, as discussed in the text.*

**Figure 2 has different fonts and size, which are partly way to small; the given precision of diameters is too high**

*It was changed as recommended.*

**Line 158: I suggest: Tir => TIR ?!**

*It was changed as recommended.*

**Line 163 a temperature is „higher" but never „warmer", same sentence: why can be a**

**relationship „straightforward" it can be clear or so...**

*Changed as recommend, the sentence was rewritten, clarifying this aspect.*

**L 165: „However, the lightning activity is highest at the same time that the aerosols have the maximum concentration, indicating that the connection between them is more complex, as will be subsequently discussed in the paper." Why indicates a relationship that the connection is complex? I cannot follow this argumentation is complex? I cannot follow this argumentation**

*The sentence was clarified.*

**L 168: again you refer in the text to a figure in the appendix – you are in the analysis section where I expect to see an analysis of your data – why is this data figure not in sec 3 ?**

*Changed as recommended, figures were included in the text.*

**L 169: I think it would be very helpful to know the accuracy of the correction for sampling losses – then you can answer the question...**

*An exact quantification is a work in progress and not finished yet. However, ATTO is now equipped with an elevator, and it will be possible to measure the concentration at the two heights, nearly simultaneously, with the same inlet.*

**Line 185: this is somewhat repetitive and can be shortened as it is classical BL theory and you have the appropriate citations already in.**

*This part of the text was shortened, the description of each stage was deleted.*

**Line 194 although I have a feeling of what you mean you should clearly define what you**

**mean with things like „relative variation"**

*It was changed as recommended.*

**Line 195 „nocturnal BL => residual layer?**

*Yes, we are referring to the nocturnal BL that has a height of around 100m.*

**Line 196ff: this argumentation is really hand waving: Do you have any information about the inversion strength and possible wind shear which determine how effective entrainment can be? Your description at this point is very speculative. Furthermore, the exchange of any scalar property strongly depends on the strength of the gradient of the scalar itself as <w'N'> scales with d<N>/dz.**

*The explanation in this line was clarified, and new references were added to the manuscript.*

**Line 198ff: *"Fog could also contribute to this behavior, though the night-time decrease is a systematic behavior during dry and wet season."* Do you have any information about fog events? Otherwise this is again pure speculation.**

*A reference was added about fog occurrence in Amazonas*

**Fig 4; it is difficult to realize which axis label belongs to which parameter – probably the right panel is shifted to far left and mask the label of the left panels?**

*The Figure position was corrected as recommend.*

**Fig 3: I am not sure if it is a convincing idea to compare the temporal evolutions of size distributions (wet & dry) with different methods (color code & isolines) – this is very irritating.**

*Both reviewers recommend splitting the former figure 3 into two figures, one for each season. We added the two suggested figures and the former one on this letter and try to convince the Reviewers that the actual figure brings additional information; it allows the reader to compare both seasons among the particles size distribution, lighting density, and brightness temperature. However, I am fully open to follow the suggestion to change to one figure for each season. In this new version, we maintained the figure in the former version.*

[Figure]

[Figure]

**Line 236/7 figure numbers are missing**

*Corrected.*

**L 249 "cloud" => "could"**

*Changed as recommend*

**Line 240ff: is there any more information of vertical wind velocity at the tower available**

**(ultrasonic anemometer?) – this could help to understand the situation in more detail – just an idea.**

*There are no micrometeorological instruments installed yet with long-term measurements. The fully equipped micrometeorological devices are being installed this year.*

**Line 243/4: The sentences do not really add to the discussion. Is it surprising that there is a correlation between updraft and lighting? - Probably not. And why is O3 and NO2**

**of any interest here? Does this help to interpret your observations? For me it looks like you want to argue that those gases trigger nucleation? Or are those gases considered as tracer for vertical motion? There is a clear red line in your interpretation missing although the observations are definitively interesting!**

*A new sentence and references were added to the text to clarify the relationship between oxidation processes and new particle formation. As a result, we believe the manuscript is now better organized and clear.*

**Maybe one idea would be to investigate one case in more detail to corroborate your hypothesis?! Concerning Fig 5, one would expect a rapid decrease of total number concentration in case of lighting events – right? Are there total counters with 1Hz-resolution available in 60 and 300 m at ATTO? I think it could be convincing to analyze such an event as a case study and then extrapolate the findings with a statistical analysis as you did.**

*This study is based on long term dataset collecting hundreds of cases in order to compute significative composites (significance tests were added as suggested by reviewer 2). We want to describe the relationship between weather events and particle size distributions, not as a case study as it was done before, but by providing statistical analysis. The opened question from this study catalyzed the installation of new instrumentation at ATTO. These new instruments will allow observing in detail, at high time resolution, with measurements of vertical motion and particle size distributions simultaneously. We are already starting to compile the data to do this, and we hope to have cases studies and a more in-depth evaluation soon.*

**Line 245: how can a volume updraft result in an increasing downward advection of ? This sentence does not make any sense to me – please verify.**

*An additional sentence was added to the text to clarify this aspect.*

**General: GLM stands for a sensor/device – right? Sometimes you use this abbreviations being a parameter (GLM density)...please specify what you really mean and be consistent.**

*It was changed as recommended*

**L 250ff: Assuming the source for high NUFP is in the upper troposphere in about 10 km or so and you have a maximum concentration observed at ground 100 min after the onset of deep convection with lightning this would imply a mean "effective" vertical downdraft of about 1.6 m/s which seems to be realistic. I suggest a few more (rough) estimates like this to see if hypotheses are at least realistic or not.**

*Thanks for, interesting consideration. We added this discussion to the manuscript.*

**Line 264: I cannot follow this argumentation, which is mainly due to the fact that Fig. 6 needs better explanation about what exactly you are doing here. And are you sure the phrase "feedback" is correct in this context?**

*We opened a new section for this part of the study. We discussed several processes from the former Figure 6, but the main feature is the difference between the seasons. The word feedback was removed, and the context better explained.*

**Line 258 : reference is missing**

*The reference was added.*

**What else do we learn more about possible processes from Fig 6 compared to Fig 5? I think the correlation between lightning and aerosol concentration is also visible in the graphs of Fig 5 (which I really like).**

*As we are focusing on the effect of clouds on the aerosol and not the opposite effect, we prefer to explore this topic further, but in fact, this feature is quite striking. We separate this result into a specific section.*

**Line 268 I am not sure if we can really should call this a "lag": with the first observed lightning we see already an increase in particle concentration and - following your argumentation – strong mixing begins. If we have diluted the lower atmosphere it is not soimportant anymore if you increase lightning/mixing as the PBL is already thoroughly mixed/diluted. Maybe I am wrong but wouldn't this also explain why the maximum in lightning is shifted to the change in aerosol concentration?**

*A discussion about these features was included in the text. And the word "lag" was replaced by time interval.*

**By the way, why not distinguishing in Fig 5 between "dry" and "wet" season as in Fig 7?**

*We have computed for wet and dry seasons separately, but we decided to add the dispersion on the figure and show all months together to describe the general processes. But in the former Figure 7, you can access the seasonal variation and the detailed discussion.*

**Line 312: such information can be shifted into the figure caption**

*It was changed as recommended.*

**Line 318: "*This observation indicates that the surface could be the source of the divergence flux of new UFP.*" What do you mean with this sentence? How can the surfaces being the source of a flux? Please clarify.**

*Thanks for the comment. The surface could be the source of UFP and not the divergence flux, and we corrected this sentence.*

**Line 325: Could dilution due to the growing well-mixed BL also be a reason?**

*We agree, and it was introduced in the text.*

**Line 333: I suggest „deep convection can produce gravity waves" – I think there need some conditions to be fulfilled for gravity waves. Furthermore, what do you mean with "vertical gravity waves"? The propagation? Please specify.**

*Changed as recommend and clarified that we are talking about vertical propagation.*

**Beginning of Sec 3.4: over seven lines you cite papers describing observations of gravity waves – but why? There is no motivation for analyzing gravity waves so far. Why are you interested in this phenomenon? This should be placed first!**

*The section name was changed to intradiurnal oscillation, and the context of gravity waves was better explained.*

**Line 340: Why applying wavelet transformation? There are some basic questions missing at the beginning of this section.**

*The section is now reformulated, and better explains the reason for the study of the intradiurnal oscillation.*

**Line 341: "…to a frequency of Tir < 284 K,…" – this is misleading, frequencies usually don't have the unit Kelvin – please rewrite this sentence**

*It was changed as recommended*

**Line 349: "..*Intracloud or intercloud variability controls convection in Amazonia and consequently the formation or reduction of particles*." I would argue that convection controls the cloud variability and not the other way around. By the way, particle formation is something different compared to increased concentration due to (vertical) transport. Did you discuss particle formation, which is usually a nucleation process? And what has this to do with gravity waves, which are the subject of this paragraph? Sorry to say but this part is a mess and needs to be completely rewritten.**

*We added an explanation of why and how gravity waves are related to the increase in ultrafine particles. We discuss two possibilities for the rise of UFP after intense rainfall events: Either by nucleation, as you mentioned, and in this case, it is possibly related to the advection of oxide gases (studies show an increase in ozone during rainfall events, a critical component in the new particles formation processes). Or the UFP is from the vertical advection of upper-level ultrafine particles. Both approaches are related to the updrafts and downdrafts that are fully controlled by intradiurnal oscillation forced by gravity waves. This context was much better explained in the manuscript.*

**Sec 3.5 starts again with an incoherent list of quotes regarding nucleation without telling the reader what this section will be about. Instead in line 365 a figure is introduced about vertical velocity observations – where is the red line in this section?**

*An introductory text was introduced in this section as recommended.*

**Line 365: „contoured frequency by altitude diagram" really does not need a shortcut.**

**The following part includes a lot of technical details that could be shifted to the technical part at the beginning of the manuscript (Sec 2.4)**

*It was changed as recommended*

**Line 370ff: Now I am confused: what exactly is the „vertical velocity w" shown in Fig 10: The velocity of raindrops or the wind velocity? The distribution is significantly skewed to negative velocities so I assume it is the drop velocity? I need to understand first what is displayed before making any comments on the scientific interpretation.**

*As mentioned in this section, the information we present is a combination of cloud dynamics and raindrops. What we want to show is that above 9 km, the dynamic is very small. We do not expect to have significant vertical motion above 10 km. The results from Die Wang et al. (2018), for GoAmazon, confirm this result. (https://doi.org/10.5194/acp-18-9121-2018). They extract the raindrop terminal velocity using a surface disdrometer and estimating the average terminal velocity using David Atlas parametrization that estimates the vertical drop velocity by the size distribution. They subtract this information as an offset and present only the vertical velocity. We didn't compute this because we just want to show that above 10 km, the cloud dynamic is reduced, and the theory that the UFP in the surface is from this layer has some limitations. This part was clarified in the manuscript.*

**Line 3745: What do you mean by this sentence? If Fig. 10 shows the droplet velocities, no statement can be made about the cloud dynamics, or am I fundamentally misunderstanding something?**

*As mentioned above,* because we want to show that above 10 km, the cloud dynamics is reduced, and the theory that the UFP in the surface is from this layer has some limitations. This part was clarified in the manuscript.

**Line 376: Which observations tell you that the maximum NUFP is above 10 km in your observational period? This paragraph does not provide a conceptional model based on your observations but rather on speculations partly based on other publications with results not necessarily valid for your situation.**

*The studies cited in this study show a high concentration of UFP in the upper troposphere. HALO results, see Andreae et al., show this feature for Amazonia. We changed the section name to Discussion and clarified what is already known, this study's contribution, and the unanswered questions. Now with the introductory paragraph at the beginning of the sections, we believe this section was clarified.*

**I don't make any detailed comments about the conclusion section as the manuscript needs major rewriting and it is not clear at this state in which direction the data interpretation will finally go.**

*We hope now the manuscript is organized as recommended*

---

## Author Comment (AC2)

**Response to Anonymous Referees**

**Article: "How weather events modify aerosol particle size distributions in the Amazon boundary layer", by Luiz A. T. Machado et al.- ACPD 2021314.**

Dear Editor,

The authors would like to thank both reviewers for their helpful comments and suggestions. Our point-to-point responses are developed hereafter, along with an indication of changes made in the revised version of the text.

. As a summary, the revisions to the manuscript include the following highlights:

- We have clarified the text. We modified the text according to the recommendations, explaining the scientific aspects and introducing each section. We also added two more subsections and renamed the former Conceptual Section to the Discussion Section. This new section was reorganized to clarify the aspects we know, the new results from this study, and the open questions.
- The gravity waves are better introduced in the text now. Thanks for the comments; the text clearly explains why gravity waves are included in the manuscript in this new version. Particle size distribution is evaluated with the diurnal, seasonal, and intradiurnal cycles. We show that the cloud intradiurnal oscillation has the same frequency oscillation as the rate of change of the particle number concentration. Further, we discuss that this intradiurnal oscillation is forced in response to the gravity waves.
- T-student tests were applied to composite studies.
- Both reviewers recommend we split the former figure 3 into two figures, one for each season. Therefore, we added in this letter the two suggested figures and the former one and try to convince the Reviewers that the actual figure brings the opportunity to compare both seasons from the particles size distribution, lighting density, and brightness temperature. However, I am fully open to follow the suggestion to change to one figure for each season. Therefore, in this new version, we maintained the figure is the former version.
- Supplementary figures were incorporated into the text.

The individual reviewer comments and responses are included in the following document (author comments in *italics*, reviewer comments in **bold**).

Sincerely,

Luiz Machado, on behalf of all co-authors

**Reviewer 2:**

**Main comments:**

**General:**

**In their paper, the authors present a combination dataset in which particle size distribution data is combined and effects of lightning and weather patterns on the aerosol particle concentrations in the Amazonas.**

*Dear Reviewer, we would like to thank your suggestions and comments, they were significant for improving and clarifying some essential aspects of the manuscript content. In the Editor's letter, we explain the main changes in the manuscript, and below, we listed these aspects related to your recommendations.*

**The dataset is interesting, and the topic and and data is certainly fitting to be published in ACP. However, I had some difficulty in fully understanding what the findings in the paper are. The conclusions first describe the typical diurnal behavior. Then, features for the dry and wet season are presented, and then features of lightining activity. This is then connected to aerosol concentrations, but the explanations of the connections are not very clear, except that higher lightning amounts seem to somewhat increase UFP number. Also, gravity waves are shortly discussed, but their role is not clarified at all. Then, downdrafts are discussed and it is noted that they could increase the UFP number. Overall, it is difficult to discern what the really new finding here is .**

*We have attempted to clarify the text. We modified the text according to the recommendations, explaining the scientific aspects and introducing each session. We also added two more subsections and renamed the former conceptual section to Discussion section and the Gravity Waves section to Intradiurnal Oscillations.*

*The gravity waves were better introduced in the text. Thanks for the comments. Now the text clearly explains why gravity waves are included in the manuscript. Particle size distributions are evaluated with the diurnal, seasonal, and intradiurnal cycles. We show that the cloud intradiurnal oscillation has the same frequency oscillation as the rate of change of the particle number concentration. Discuss that this intradiurnal oscillation is forced in response to the gravity waves.*

*Our main results present statistical results showing how the diurnal cycle, the seasonal cycle, and the intradiurnal oscillations modify particle size distribution. In addition, we describe how different cloud characteristics are associated with the PSD. Finally, we present the particle size distribution changes during lighting events, but we also show how particle size distribution changes as functions of the frequency of clouds with high cloud tops, VIL, and rainfall.*

**The manuscript also presents a section named conceptual model, but to me this seemed mostly to be a review of previous literature and the connection to the present dataset is not very strong. The analysis here seems to show that there are three different periods: wet season, dry season, and transitions season, during which lightning activity is strongest. Would it be possible to show a clear model, for example**

**as a figure, that shows for each season what kind of phenomena are the most important ones affecting the PSD during each season in light of the data? This could then be discussed with the supporting data, and the open questions remaining .**

*The new Discussion section (replaced the former Conceptual model) was reorganized to clarify the aspects we know, the new results from this study, and the open questions.*

**Finally, the authors state that the dataset opens up new scientific questions, but they do not elaborate what these questions are. As of now, the paper reads like a report on data analysis, where interesting features are found, but the critical analysis of these features is missing .**

*We reorganized the manuscript, added new subsection and an introductory paragraph in each section. As a result, we believe the manuscript is better organized and understandable.*

**Based on this, I would suggest a major revision of the text that clarifies the novel findings - for example, instead of weather events, why not directly discuss strong lightning events? Also, more strongly present an argument of what is happening, and how the data supports this interpretation. If new questions arise, they could be stated along with the reason why they cannot be answered with the current dataset .**

*The study also presents how particle size distribution change as a function of cloud top, rainfall, and cloud liquid water. Therefore we consider that this study covers more than only lighting. The manuscript was reorganized, and these aspects were clarified. We study how particle size distribution changes as functions of the diurnal, season cycles and intradiurnal oscillation, the main driver of the cloud variability in Amazonia. Besides, we look at how cloud characteristics are associated with particle size distribution and how weather events modify particle size distribution. We believe this reorganization of the manuscript clarifies all these points raised by the Reviewer.*

**Some additional comments:**

**- Figure 3: In my opinion, using a different plotting scheme for dry and wet season inside the same figure is highly confusing. Two similar figures make comparisons possible .**

*Both reviewers recommend splitting the former figure 3 into two figures, one for each season. We added on this document the two suggested figures and the former one and try to convince the Reviewers that the actual figure brings additional information, it allows the reader to compare both seasons among the particles size distribution as well as from lighting density and brightness temperature. However, I am fully open to follow the suggestion to change to one figure for each season. In this new version, we maintained the figure in the former version.*

[Figure]

[Figure]

[Figure]

**- Diurnal cycle: The figures in the appendix seem very contradictory to the presented diurnal cycle. In the conclusions, a diurnal cycle where UFP has a maximum at sunrise and Aitken and accumulation particles have a minimum is presented. However, the averaged diurnal cycles in Figs A3 and A4 are very different from this, as the maxima and minima seem to occur around 10-12 o'clock, much later than the sunrise. This should be clearly clarified, as such data interpretation seems very strange .**

*These figures were added to the main text. The time is in UTC, LST – 4, we comment on this feature in the text, but we decided to add to the legend to clarify this point.*

**- Statistical testing: In many cases, the differences between cases seems to be a small signal in the number concentration data. Therefore, statistical testing is important to show that the signal is not just chance.For example, figure 5 seems bring a clear distinction to the particle concentration data, and it is also very interesting. The difference between low amount of lightning and high amount of lightning seems significant, but would it be possible to statistically test whether the shown data is also statistically significant? I assume it is, based on the high number of measurements, but it would be good to have an analysis on this. The same can be said for Fig. 6: I think here, it is essential to show via statistical testing whether there is any significant difference between the particle numbers when the GLM number changes. I think quite straightforward tests should be applicable here.**

*T-student statistical tests of significance were applied to the composites, and the results are now commented in the text.*

---

## Author Response (AR2)

**Article: How weather events modify aerosol particle size distributions in the Amazon boundary layer by Luiz A. T. Machado et al.- ACPD 2021314.**

Dear Editor Prof. Ken Carslaw,

The authors would like to thank the reviewers for their comments and suggestions. We have changed the manuscript accordingly to the reviewer and Editor's comments. Our point-to-point responses follow below:

1) Fig 3c and 3d: If UTC is used for time, indicating local noon on the figure would be very helpful to get an idea of the variation.

*Changed as suggested, the Noontime is indicated in both Figures.*

2) Fig 4: I still consider the separate figures much better. For example, the bi-modal nature of the PSD in the wet season becomes clearly apparent in the new version. I suggest using the new version (with legends corrected for the wet season). The increased opportunity for intercomparison is at least in my mind completely lost to the irritation of trying to understand what the lines are about.

*I couldn't convince the reviewer that the combined figure is better, so I changed it to two figures as suggested.*

3) I also recommend another careful language check and fix typos, e.g.: Line. 161: frequency-b- altitude -> frequency-by-altitude. Line 464 "... Wang et al. (2018) compute" -> computed. Line 476 "...Pleiades of processes" -> maybe you mean a myriad or something similar?

The manuscript was carefully checked again, and the correction was done as recommended.

4) Comment from the Editor - I would also like to see some clarification on the altitude range over which you believe the downdrafts are occurring. In a few places, it reads as if you think that downdrafts can travel from an altitude of 10-14 km down to the surface. The energetics of that (in terms of potential temperature) would need to be explained. I appreciate that cloud dynamics is not the topic of your paper, but such implied air motion as an explanation for your aerosol observations should be energetically plausible. There are many places in the paper where you are quite vague about the altitude range of downdrafts, so please carefully review your assertions or suggestions.

I am sorry if it reads as downdrafts travels from the top of the upper troposphere in some parts of the text. This hypothesis makes no sense, and I show the radar wind profile statistics to demonstrate that the vertical motion is behind this layer. Many papers mention this downward transport from this 14-10 km layer, and it has no reason as the downdrafts are much lower. I read the manuscript again and clarified the part of the text where could have a different interpretation.

---

## Author Response (AR3)

**Article: How weather events modify aerosol particle size distributions in the Amazon boundary layer by Luiz A. T. Machado et al.- ACPD 2021314.**

Dear Editor Prof. Ken Carslaw,

Thanks for helping make this sentence more exact!
I hope now I have clarified what I was trying to say; please see the text below I changed in the manuscript in the Conclusions:

Finally, the vertical distribution of the vertical velocity was computed using the radar wind profiler at the ATTO-Campina site. The data were collected in the season of the most intense convective activity (October – December) and show that downdrafts are mainly located below 10 km, while the layer of maximum concentration of UFP is mainly above 10 km. However, if even a fraction of UFP from the middle levels of the troposphere is advected downward, it could support the concentration of UFP in the surface observed during convective events. The NUFP in the upper troposphere is several orders of magnitude higher than at the surface, but in the middle level of the atmosphere, there is still have a concentration of UFP larger than on the surface. The source of the UFP at the surface during convective events is still an open question because there is also the mechanism of particle formation related to the increase of OH and NOx during convective events.